



# Transport model diagnosis of the mean age of air derived from stratospheric samples in the tropics

Hanh T. Nguyen[1], Kentaro Ishijima[2,*], Satoshi Sugawara[3], and Fumio Hasebe[4]

[1]Graduate School of Environmental Science, Hokkaido University, Sapporo 060-0810, Japan
[2]Japan Agency for Marine-Earth Science and Technology, Yokohama 236-0001, Japan
[3]Miyagi University of Education, Sendai 980-0845, Japan
[4]Faculty of Environmental Earth Science, Hokkaido University, Sapporo 060-0810, Japan
[*]Present affiliation: Meteorological Research Institute, Tsukuba 305-0052, Japan

**Correspondence:** Hanh T. Nguyen (hanhnt@ees.hokudai.ac.jp)

**Abstract.** Stratospheric profiles of the mean age of air estimated from cryogenic air samples acquired during the CUBE/Biak field campaign over Indonesia are investigated with the aid of an atmospheric chemistry transport model nudged to ERA-Interim meteorological fields. Application of the boundary impulse response (BIR) method and Lagrangian backward trajectories to the transport field simulated by a single model prove useful in interpreting the observational results, which include

discrepancies between $CO_2$- and $SF_6$-derived mean ages. This may be because the BIR method takes unresolved diffusive processes into account while the Lagrangian method distinguishes the pathways the air parcels have taken before reaching the sample site. The capability to estimate the vertical profiles of the clock tracer concentrations and the water vapor "tape recorder" is another advantage of the Lagrangian method, confirming the reality of the trajectory calculations. The profile of $CO_2$-mean age is reproduced reasonably well by trajectory-derived mean age, while BIR-derived mean age is much greater than $CO_2$ age

at 28 and 29 km, possibly due to high diffusivity in the transport model. On the other hand, $SF_6$ age is reproducible only in the lower stratosphere, but far exceeds the trajectory-derived mean age above 25 km. As air parcels of mesospheric origin are missing in the Lagrangian age estimation, this discrepancy, together with the fact that the observed $SF_6$ concentrations are much lower than the trajectory-derived values in this height region, is consistent with the idea that the stratospheric air samples are mixed with $SF_6$-depleted mesospheric air, leading to overestimation of the mean age.

## 1  Introduction

The tropical atmosphere is characterized by the highest and coldest tropopause among others as well as warm and humid air in the lower troposphere. In a study of stratospheric dryness, Brewer (1949) noted these features as the key to understanding the global-scale stratospheric circulation. His water vapor observations, along with the interpretation of the global ozone distributions by Dobson (1956), led to the conclusion that the tropospheric air enters the stratosphere primarily through the

cold tropical tropopause and spreads toward high latitudes. This Brewer–Dobson circulation (BDC) constitutes the basis of the dynamics as well as tracer transport in the stratosphere. The depletion of stratospheric ozone (e.g., Crutzen, 1972; Molina and Rowland, 1974; Farman et al., 1985) proved that the stratosphere is vulnerable to anthropogenic forcing. However, this





forcing causes environmental issues other than just ozone depletion. Changes in stratospheric water vapor can affect the global climate by driving decadal-scale fluctuations in global mean temperature, even though the mixing ratio is as low as $\sim 4$ ppmv (Solomon et al., 2010).

The response of the BDC to natural and anthropogenic forcings is one of the central issues of current stratospheric research
(Hasebe et al., 2018, hereafter referred to as H18). Modulation of the BDC, such as changes to the velocities and pathways of circulation, is often diagnosed by stratospheric age of air (AoA), which is defined as the time a tropospheric air parcel has spent in the stratosphere since its entry (Kida, 1983). The mean value of the AoA (mean age) can be estimated by successive observations of clock tracers (Schoeberl et al., 2005; Ploeger et al., 2015) such as carbon dioxide ($CO_2$) and sulphur hexafluoride ($SF_6$). These constituents are chemically stable while their concentrations increase monotonically in the troposphere, which makes it
possible to use their stratospheric mixing ratio as the time stamp of their entry to the stratosphere. Accumulated observations of clock tracer concentrations in the northern midlatitude stratosphere (Nakazawa et al., 1995; Andrews et al., 2001a; Nakazawa et al., 2002; Aoki et al., 2003) have been used to estimate the long-term trend of mean age (Engel et al., 2009). The results show a small increase at a rate of $0.24 \pm 0.22$ yr decade$^{-1}$, although the statistical significance is limited to the 68 % level. However, these results are not consistent with the long-term decrease of mean age diagnosed by chemistry–climate models (e.g., Austin
et al., 2007; Garcia and Randel, 2008). There are uncertainties in both observational and model estimates, including limited observational data of seasonally varying $CO_2$ concentration and unavoidable use of parameterization schemes associated with coarse model resolution (e.g., Garcia et al., 2011; Stiller et al., 2012; Diallo et al., 2012). Until now, the discrepancy between observation and model has remained a subject of debate. Stratospheric sampling of clock tracers was conducted during the CUBE/Biak campaign (H18) in Indonesia at the entry region for the air to the stratosphere with the aim of avoiding, as much
as possible, complexities arising from multiple pathways and unresolvable diffusion. The present study is one of our efforts to understand the CUBE/Biak observations.

The conventional method of age estimation relies on the so-called lag time method (Elkins et al., 1996; Bischof et al., 1985; Boering et al., 1996; Harnisch et al., 1996; Patra et al., 1997; Hall and Plumb, 1994; Andrews et al., 2001a; Waugh and Hall, 2002; Haenel et al., 2015), which uses the delay of the time series in the stratosphere relative to that in the troposphere. However,
the simple use of this method can introduce appreciable errors if the time series is not a monotonically increasing function (Waugh and Hall, 2002). An alternative method is to interpret the stratospheric concentration as a response to tropospheric forcing (Hall and Plumb, 1994; Waugh and Hall, 2002). Any stratospheric air parcel can be decomposed into small fluid elements or particles, each of which has its own transport pathway and different transit time. The age of a single particle does not represent the atmospheric transport processes (Hall and Waugh, 1997; Schoeberl et al., 2005). Instead, the age spectrum
(Kida, 1983; Hall and Plumb, 1994), which is the probability distribution function (pdf) of the transit times of all these particles, is used to specify the air parcel statistically. Unlike the mean age, however, the age spectrum cannot be obtained directly from observed data (Andrews et al., 1999, 2001b; Johnson et al., 1999; Ray et al., 2017).

Estimation of the age spectrum was made possible by using the pulse tracer method introduced by Hall and Plumb (1994). Under this method, tracers are released in a chemistry transport model (CTM) under steady flow conditions using the forward
tracer equation (Eulerian approach). The steady flow constraint is overcome by employing a large number of artificial impulse





tracers to take account of time-varying flow in estimating the distribution of transit time (Haine et al., 2008). The mixing ratio of an impulse tracer as a function of time at any given stratospheric point is considered as a response to a boundary source region $\Omega$. This method is known as the boundary impulse response (BIR). The age spectrum is derived from the BIR map and the mean age is the first moment of the age spectrum (Haine et al., 2008; Li et al., 2012a). Alternatively, the age spectrum

can be estimated by following the original Lagrangian viewpoint. H18 estimated the age spectrum by backward trajectory calculations and sought to reproduce the vertical profiles of the observed $CO_2$ and $SF_6$ mole fractions, mean ages, and water vapor mixing ratio simultaneously.

In the present study, the BIR and Lagrangian methods are employed with the aid of an atmospheric chemistry transport model (ACTM) to derive the age spectra in the tropical stratosphere, in order to understand the vertical profiles of $CO_2$ and

$SF_6$ mole fractions and mean AoA observationally estimated by Sugawara et al. (2018) (hereafter referred to as S18). Trajectory calculations of H18 are updated by extending the integration period by a factor of 3 and using 4 times the number of trajectories as those in H18 to improve the credibility of the results. The ACTM model and BIR methods are described in Sect. 2, the results of estimating the mean age are presented in Sect. 3, and the differences in the estimated mean age profiles are discussed in Sect. 4. Finally, a summary is given in Sect. 5.

## 2  Model and experiment

### 2.1  Atmospheric chemistry transport model

The present analysis employs an on-line ACTM based on the Center for Climate System Research/National Institute for Environmental Studies/Frontier Research Center for Global Change (CCSR/NIES/FRCGC) atmospheric general circulation model (AGCM) (Numaguti et al., 1997). It is configured to have 67 sigma levels from the surface to a height of approximately 90 km.

The horizontal resolution is T42 spectral truncation (equivalent to $\sim 2.8° \times 2.8°$, Ishijima et al., 2010). The cumulus convection parameterization scheme is the simplified Arakawa and Schubert scheme (Arakawa and Schubert, 1974; Numaguti et al., 1997). The adjusted cloud mass flux is used to calculate the updraft and downdraft due to cumulus convection. For the calculation of tracer transport, ACTM applies the Piecewise Parabolic Method scheme with a monotonic constraint (a type of van Leer scheme) and a flux-form semi-Lagrangian scheme (Lin and Rood, 1996). The second-order vertical eddy diffusion scheme of

Mellor and Yamada with cloud effects (Mellor and Yamada, 1982; Numaguti et al., 1997) is applied for subgrid-scale vertical fluxes of meridional velocity, temperature, surface pressure, and mixing ratios of tracers. The model has been evaluated in an AoA inter-comparison with other transport models (Krol et al., 2018), and has been used in studies of the transport properties of chemical constituents such as $N_2O$ (Ishijima et al., 2010) and in demonstrating the utility of a novel three-dimensional transport formulation (Kinoshita et al., 2019).



## 2.2 Simulation design

Tracer transport calculations using the ACTM have been conducted for the period from January 2005 to March 2015. The initial meteorological fields on 01 January 2005 were produced by spinning up the ACTM meteorology, nudged towards the horizontal winds and temperature of the European Centre for Medium-Range Weather Forecasts Interim Reanalysis (ERA-

Interim; Dee et al., 2011) at 6-hour time intervals for several years. The effect of data assimilation to the transport features is investigated by comparing a free run (no nudging) of the ACTM (AF) with the ERA-Interim (EI) nudged run (AN). For BIR calculations, pulse tracers are released from the source region $\Omega$ placed at the tropical surface corresponding to the latitude band $15°$ S–$15°$ N by setting the mixing ratio to a constant value (1 ppbv) for a month and to zero otherwise, following Ploeger and Birner (2016). The releases are made in odd-numbered months; i.e., the source time is set to January, March, May, July,

September, and November of each year throughout the simulation period (January 2005 to March 2015), resulting in the release of 62 distinct tracers in total.

## 2.3 Simulated tracer fields

The simulated transport features are illustrated by the monthly mean zonal mean values averaged over nine pulse tracer fields released on the same calendar month but in different years from 2005 to 2013. Considering the enhancement of poleward

transport in winter, we examine the distribution and the local rate of change of pulse tracers during the months following the releases in January and July to describe the stratospheric transport features in particular. Figure 1 shows the latitude–height distributions of the zonally averaged monthly mean tracer concentrations (ppbv) in February (top) at (a) the 2nd month and (b) 14th month from release in January. The bottom panels are the corresponding distributions in August for July-released pulse tracers. Contours of the Transformed Eulerian Mean (TEM) residual stream function averaged for the same 9 year period using

ERA-Interim data are shown in white. During the few months after release, the tracer is transported by the tropospheric Hadley circulation from the source region vertically to the upper troposphere and horizontally toward high latitudes. The deflection to the summer hemisphere of the ascending branch of the Hadley circulation traps the tracers primarily inside the winter hemispheric Hadley cell (Fig. 1(a, c)), although some tracers are pumped up into the TTL and the lower stratosphere (LS). In the TTL/LS region, there is higher concentration in the winter hemisphere, suggesting a stronger circulation in wintertime. One

25  year later, the surface-released tracers have been transported into the stratosphere, particularly in the tropics. The stratospheric concentrations are smaller and the transport speed is slower for the release in July (bottom panels) than in January (top panels). This can be understood in terms of the hemispheric asymmetry in the strength of the stratospheric wave-pumping during winter. Of note, we see a distinct maximum in the tropics (500 K isentrope $\sim 50$ hPa), especially in February (Fig. 1(b)) in spite of the widespread TTL maximum one year earlier (Fig. 1(a)).

To understand the transport features responsible for redistributing the tracers, the continuity equation for the pulse tracer is considered. The local rate of change of the zonal mean tracer mixing ratio $\overline{\chi}$, is described by the zonally averaged continuity equation expressed as follows in the TEM framework (Andrews et al., 1987, Eq. (9.4.13))

$$\frac{\partial \overline{\chi}}{\partial t} = -\overline{v}^* \frac{\partial \overline{\chi}}{\partial y} - \overline{w}^* \frac{\partial \overline{\chi}}{\partial z} + \frac{1}{\rho_0} \nabla \cdot \boldsymbol{M} + P - L. \tag{1}$$





where $(\overline{v}^*, \overline{w}^*)$ are the TEM residual velocities, the overbar indicates a zonal mean, $M$ is the TEM eddy flux vector, and $P$ and $L$ are the chemical sources and sinks, respectively.

Figure 2 shows the tendency of the tracer mixing ratio during the one-month period from the 14th month (February and August) to 15th month (March and September) after release (in January and July, respectively). Panels (a, e) show the total

change (left-hand side of Eq. (1)), while those in (b, f) and (c, g) show the contribution from the first and the second terms, respectively, of the right-hand side of this equation. Panels (d, h) show the residual of all the other terms, corresponding to the sum of eddy transport, unresolved small-scale mixing, and source/sink terms. The total tendency appears positive (negative) above (below) the $\sim 500$ K isentrope except at high latitudes. The stratospheric tendency has a maximum in the tropics of the summer hemisphere due partly to the vertical advection, whereas the widespread increase in the extratropics is driven mainly

by the horizontal transport from the tropics. The decrease in the summer polar latitudes is due to poleward and downward advection. In the extratropical LS, the contribution of the residual terms (probably eddy mixing) appears to dominate. Of note, during winter the contribution of the horizontal advection in the LS is negative in the Northern Hemisphere (NH, Fig. 2(b)) but positive in the Southern Hemisphere (SH, Fig. 2(f)).

## 2.4   Boundary Impulse Responses and age spectra

The transport features shown above are limited to those of selected tracers at a specific time from release. To assess the time evolution of tracer fields, examples of BIRs are investigated. Figure 3 shows the January (red) and July (blue) BIRs expressed in the form of a pdf at 100 hPa, 50 hPa and 10 hPa over the equator (the average of two BIRs at $1.4°$ N and S) and at latitudes $20.9°$ N, $46.0°$ N and $79.5°$ N at 50 hPa. Since our ACTM transport calculations cover the period from January 2005 to March 2015, five BIRs with source times from January 2005 to November 2009 can be averaged by terminating the transit time at five

20   years. The results are shown by the thick lines with the five-year range between maxima and minima shown by shading. In the tropics, the BIR-peaks appear earlier and higher for the tracers released in January than in July on 100 hPa. However, this is not always the case on the 50 and 10 hPa levels due to modulation of the ascending velocity by the quasi-biennial oscillation (QBO), as discussed in Sect. 3.1. The BIR at 50 hPa over the equator reaches a maximum several months after release. At 10 hPa, the transit time of about two years necessary for the surface tracers to reach a maximum agrees well with that in Li

et al. (2012b) (Fig. 1(a)), although the decay time appears longer in the present study. The latitudinal variation of the BIRs on 50 hPa shows a tendency for peaks to be later and smaller with longer decay time as we go to higher latitudes. The seasonal dependency of our results at $20.9°$ N on 50 hPa agrees with that at $20°$ N on the 420 K isentrope ($\sim 70$ hPa) in Li et al. (2012a). That is, the January-BIR shows an earlier and higher peak than the July-BIR.

The transverse section of the BIR map at a specific time and location constitutes the age spectrum of the corresponding

location at a fixed field time (e.g., Haine et al., 2008; Li et al., 2012a). Figure 4 shows the 3-year (2012–2014) mean age spectra over the equator and $79.5°$ N and S in the lower stratosphere. The age spectra (black curves) obtained at the equator and at high latitudes show annual multiple peaks, with long tails even in the tropical stratosphere, consistent with previous studies (e.g., Reithmeier et al., 2008; Li et al., 2012a; Ploeger and Birner, 2016). These age spectra are used to calculate the mean age $\Gamma$, the first moment of the age spectrum, by integrating from zero to infinity. However, the termination of the transport





calculations means that the transit time becomes shorter for tracers launched later in the simulation period. To cope with this problem, a tail correction has been applied in previous studies (e.g., Hall et al., 1999; Reithmeier et al., 2008; Li et al., 2012a; Ploeger and Birner, 2016). Following Ploeger and Birner (2016), an exponential function is fitted to each age spectrum derived from transport calculations over 3 years (from the fourth to seventh years of the transit time shown by the dashed line) and

the decay rate obtained is used to extrapolate the age spectrum to infinity (solid red line). The tail correction appears more significant with increasing latitude and altitude. In January, the corrected mean age, $\Gamma_{\text{corr}}$ (vertical red line), is several weeks longer than the uncorrected value ($\Gamma^*$, in black) at 50 hPa over the equator and more than a half year longer at 100 hPa at 79.5° N. In the high-latitude stratosphere, both $\Gamma^*$ and $\Gamma_{\text{corr}}$ are longer in January than in July. The modal time (spectral peak), which is independent of the tail correction, also appears later in January than in July, in spite of the earlier emergence of BIR

peaks in January than in July in the equatorial stratosphere at 50 hPa (Fig. 3). These features suggest that the seasonality of the deep branch of the BDC, especially the sinking portion at high latitudes, behaves somewhat differently from that of the equatorial lower stratosphere.

To evaluate the mean age estimation by our ACTM, the latitude–height section of the seasonal mean $\Gamma_{\text{corr}}$ is shown in Fig. 5 with the potential temperature as the vertical coordinate. Since our simulation design, the method of tail correction, and the

use of ERA-Interim meteorological fields follow those of Ploeger and Birner (2016), it is possible to compare our Fig. 5 with their Fig. 4. There is good agreement overall in features including the seasonal asymmetry between the hemispheres, strong latitudinal gradient associated with the subtropical mixing barriers, small slope of contours in high-latitude LS in the summer hemisphere, and so on. However, there are some quantitative differences at high latitudes. In the case of December-January-February (DJF) average, for example, the 4.5 year contour lies above the 650 K and 500 K surfaces at 90° S and 90° N,

respectively, in Fig. 5, while it lies below 500 K and 450 K, respectively, in Ploeger and Birner (2016), indicating that the mean age is younger by about 0.5 to 1.0 year in our ACTM. This tendency also appears in June-July-August (JJA) average, although the difference is less evident. These features, suggesting an incomplete isolation of the air inside and outside of the polar vortex, are thought to have arisen mostly from slightly too diffusive transport of the ACTM, due mainly to the relatively low horizontal resolution (2.8° × 2.8°). In spite of the availability of only 3 years (2012–2014) for estimating the multi-year

age spectra, the overall agreement is encouraging. In the next section, we derive the age spectra and mean age profile for the air parcels corresponding to the cryogenic air samples taken by CUBE/Biak.

## 3   Application to the tropical stratosphere

Stratospheric tracer transport is characterized by the superposition of a rapid quasi-isentropic mixing and slow diabatic overturning circulation (e.g. Haynes et al., 1991; Holton et al., 1995). The subtropical mixing barrier partially protects air parcels

inside the "tropical pipe" (Plumb, 1996) from eddy mixing. The conceptual view of the "leaky pipe" model, extended to include detrainment of surf zone air into the tropics (Neu and Plumb, 1999), makes the argument of tracer transport in the tropical stratosphere more comprehensible. The tropical stratosphere is also characterized by the availability of the water vapor "tape recorder" (Mote et al., 1996), in which water vapor anomalies resulting from seasonally varying tropopause temperature are



imprinted as stripes. This provides an independent measure of the ascending motion and the transit time of air parcels in the stratosphere, although the mean age and the water vapor phase lag should not be directly compared (Waugh and Hall, 2002). The CUBE/Biak field campaign was conducted to take advantage of these tropical processes in the study of the stratospheric circulation (H18). From the detailed analysis of collected air samples, S18 described the vertical profiles of $CO_2$ and $SF_6$ ages

from the TTL to 29 km. In this section, we seek to assess the observationally estimated mean ages by employing two methods, the BIR method and the back trajectories, to derive the vertical distributions of the age spectra as well as the mean age.

### 3.1   Boundary Impulse Response method

One of the advantages of the BIR method is its capability to estimate non-stationary age spectra for complete three-dimensional transport including mixing (Ploeger and Birner, 2016). On the other hand, caution is required because the results are strongly

dependent on the performance of the transport model used. The transport features shown in the previous section demonstrate that the ACTM nudged to ERA-Interim does a reasonably good job, enabling us to make a quantitative assessment of the age profile in the tropical stratosphere. Before making direct comparisons, however, it is necessary to address the difference in the definition of the mean age. That is, the source region $\Omega$ of our BIR method is assigned to the tropical surface, while the observational estimates of the mean age by S18 refer to the tracer concentrations in the tropical upper troposphere following the

original definition by Kida (1983). As mentioned in Sect. 2, the stratospheric AoA depends on tropospheric transport features including isentropic mixing with the air in the extratropical LS, and thus the relationship between the mean ages counted from the tropical surface and from the TTL is not simple. On the other hand, assigning the source region to the TTL is found to create an alternative difficulty, since the TTL covers a wide range in latitude and height, making it inappropriate to regard it as a single source region. To avoid complexity, we use the adjusted mean age $\Gamma_{adj}$ defined by subtracting the mean transit time

at the top of the troposphere ($Tr_{top}$) from the $\Gamma_{corr}$ estimated by the BIR method for the comparison with the observational estimates. In this analysis, $Tr_{top}$ is taken to be the 355 K isentrope as in H18 and the latitudinal averaging is applied between 30° N and S.

The BIR map at 50 hPa over the equator built from all 62 pulse tracers released during the period from January 2005 to March 2015 is shown in Fig. 6(a). Tracer concentrations tend to be high during boreal winter. In addition, there are intermittent

higher peaks (viewed as orange and red) identifiable, for example, at the (source time, field time) coordinates around (March 2007, October 2007) and (November 2009, July 2010). In these cases, the transit zone of the pulse tracers, that is, the tropical lower stratosphere below 50 hPa, is mostly covered by easterly shear in the zonal wind (Fig. 6(b)). This means that the secondary circulation associated with the equatorial QBO (Plumb and Bell, 1982; Baldwin et al., 2001) drives the tracers upward. Thus, as discussed by Ploeger and Birner (2016), the higher peaks of the BIR map occur where the QBO facilitates

seasonal intensification of the wave-driven pumping from the northern midlatitude stratosphere.

Figure 7 shows the meridional section of $\Gamma_{corr}$ in March 2015 estimated by the BIR method. The vertical axis is taken as altitude for comparison with the cryogenic air samples. The latitudinal split, centred over the equator at around 25 km and extended toward 30 km with a deflection to the NH, is due to the perturbation by the downward motion associated with the westerly shear of the QBO (Fig. 6(b)). The locations of the $Tr_{top}$ (355 K) and the top of the TTL (400 K) are marked by white



line segments between 30° N and S. The mean transit time at the level of $Tr_{top}$ varies with latitude in the range $0.68 \pm 0.69$ years (mean $\pm$ standard deviation). Then, $\Gamma_{adj} \equiv \Gamma_{corr} - 0.68$ years is regarded as the stratospheric mean age estimated from the BIR method for the rest of this paper.

Figure 8 shows the age spectra at eight altitudes corresponding to CUBE/Biak cryogenic air samples. These are used to estimate not only $\Gamma_{corr}$ but also $\Delta$, the spread of transit times (Hall and Plumb, 1994; Waugh and Hall, 2002), as the estimates of uncertainties in $\Gamma_{corr}$. The vertical distributions of mean age will be discussed in Sect. 4 by comparison with those estimated by the Lagrangian method (Sect. 3.2) and derived from CUBE/Biak air samples (S18).

## 3.2    Lagrangian method

A bundle of kinematic backward trajectories is calculated to estimate the spectra of stratospheric AoA, $CO_2$ and $SF_6$ mole
fractions, and water vapor mixing ratio by tracking the position of air parcels advected by the three-dimensional wind. The method of analysis is the same as that presented in Hasebe and Noguchi (2016) and Hasebe et al. (2018). The experimental conditions are summarized in Table 1. ACTM nudging simulations make it possible to employ ERA-Interim meteorological fields at higher time resolution with noise-reduced averaged (rather than instantaneous) values to better describe realistic transport. The pressure levels additional to the ERA-Interim configuration in the TTL help to better resolve the Lagrangian
cold point (LCP), which is critical to the precise estimation of the "tape recorder" profile.

The air parcel advected along each trajectory is assumed to retain the mole fractions of $CO_2$ and $SF_6$ at the time of the last passage through the $Tr_{top}$ and the saturation water mixing ratio (SMR) that takes a minimum ($SMR_{min}$) at the LCP. The specific values for a bundle of backward kinematic trajectories initialized at the time and place corresponding to each cryogenic air sample constitute the spectra of $CO_2$ and $SF_6$ mole fractions, and water mixing ratio. The age spectrum is estimated by counting
the time since the last passage through the $Tr_{top}$ for each trajectory. The $CO_2$ and $SF_6$ mole fractions at the $Tr_{top}$ are taken from the CONTRAIL data compiled by S18. Figure 9 shows an example of such spectra estimated in this way for Sample 8 of the CUBE/Biak observations with the AoA and $SMR_{min}$ from left to right at the top, and $CO_2$ and $SF_6$ mole fractions at the bottom. In this calculation, all air parcels that cross the $Tr_{top}$ transported by AN wind field are used. The number of air parcels amounts to 8272 out of the 8559 released in total; uncounted air parcels either stay in the stratosphere for 10 years or have
reached the top boundary of the calculation domain (1 hPa). The age spectrum has a long tail, exceeding 2000 days, which is responsible for creating a long tail in the low concentration (left-hand) side of the $CO_2$ and $SF_6$ mole fraction spectra. Due to the strong seasonal variations superposed on the long-term increase in the troposphere, the $CO_2$ spectrum has a more complex shape than that of $SF_6$. In contrast, the water spectrum has a single peak as the spectral shape is linked to the seasonal cycle of TTL temperature.

The tail correction for the estimation of mean age is applied in a similar manner to that used for the BIR method. In the case of Sample 8 (Fig. 9), $\Gamma_{corr}$ is 794 days, which is only 18 days longer than $\Gamma^*$ of 778 days. The tail corrections for other air samples are smaller than that of Sample 8, and we can thus safely conclude that our 10-year trajectory calculations are long enough to estimate the mean age, and that the tail correction is not critically important in the present analysis of the tropical stratosphere using back trajectories.





Vertical profiles of mole fractions of $CO_2$ and $SF_6$, $\Gamma_{corr}$, and water mixing ratio corresponding to the CUBE/Biak campaign are calculated. The results estimated using the ERA-Interim (EI) reanalysis field (not shown) remain almost the same as in Fig. 11 of H18, indicating that the truncation ($\leq 1200$ days) of age spectra is not the main reason for the deviations from observations. Figure 10 shows the results from the present study that relies on the meteorological field nudged towards ERA-Interim reanalysis (AN). The profile of $\Gamma_{corr}$, to be shown later in Sect. 4, is omitted here. In this calculation, only the troposphere-to-stratosphere transport (TST) trajectories, defined as those traceable down to 340 K isentrope recording an $SMR_{min}$ in the TTL, are used. There is a remarkable improvement on H18 in the sense of similarity to the observational estimates. The profile of $CO_2$ mole fraction is reasonably well reproduced, including the rapid decrease between 18 and 25 km and near constant values above. The improvement is also confirmed for $SF_6$ below 24 km. Above 25 km, however, the improvement is limited and the estimated $SF_6$ mole fractions remain much larger than those observed. These features are discussed in Sect. 4. In contrast to AN runs, ACTM free (AF) runs provide unrealistic values (not shown), indicating that the vertical advection is too rapid.

The results shown above indicate that the realism of the estimated profiles strongly depends on the mass transport, especially in the vertical direction. Figure 11 compares snapshots of the meridional location of air parcels corresponding to Sample 8 calculated by backward trajectories using EI, AN, and AF runs on 1 January 2015 and 1 June 2014 (57 days and 271 days from initialization, respectively). During the NH winter (January and February 2015), air parcels, colour-coded by potential temperature, gradually descend in reverse time sequence, and stay mostly inside the "tropical pipe" without appreciable latitudinal dispersion. The anisotropic displacement to the SH is related to the deflection to the summer hemisphere of the ascending branch of the BDC in the tropical LS. The vertical advection is fastest in AF and slowest in AN. It is interesting that EI and AN differ, even though the wind field in AN is nudged to that of EI. By 1 June 2014 (lower panels), an appreciable portion of AF air parcels has descended back to the troposphere, with some parcels migrating adiabatically on the isentropes near the bottom of the stratosphere. Most of the stratospheric air parcels with potential temperature $> 400$ K are found in the SH. In contrast, scarcely any air parcels have descended to the troposphere in AN. EI shows features intermediate between AN and AF. The difference in the vertical advection depicted in Fig. 11 must be directly coupled to the spectra of the AoA and the mole fractions of $CO_2$ and $SF_6$. It must have been transferred also to $SMR_{min}$ through the seasonal dependence of LCP temperature.

Why is the advection velocity in EI and AN different in spite of nudging? Figure 12 compares the time series of three-dimensional wind components at $0°$ longitude and 70 hPa over the equator. The horizontal wind components of AN mostly follow EI winds, as expected. However, there is some tendency for the vertical wind fluctuations to be smaller in AN than in EI, possibly due to the dynamical constraints of the AGCM. The time evolution in one-hour intervals combined with the noise reduction by averaging for an hour in AN may be the reason for the slower (and probably more realistic) vertical displacement of kinematic trajectories. On the other hand, the amplitude of variations is much larger in the AF run than in the AN run and EI analysis for both horizontal and vertical wind components. The variations in vertical wind appear rather noisy, causing more rapid and possibly spurious vertical displacement of air parcels.





## 4 Discussion

We have employed two methods, the BIR method and back trajectory calculation, to estimate the mean age profile corresponding to the CUBE/Biak air samples. Comparisons of these results against those estimated by using $CO_2$ and $SF_6$ as clock tracers (S18) provide useful information on the validity and limitations of the estimation using both ACTM and clock tracers. To avoid

confusion when making comparisons between mean ages estimated from different methods, in this section $\Gamma_{adj}$ estimated by the BIR method is denoted by $\Gamma_{bir}$, the $\Gamma_{corr}$ obtained by trajectory calculations (no adjustment required) is expressed by $\Gamma_{trj}$, and the observational estimates of the mean age from $CO_2$ and $SF_6$ samples are written as $\Gamma_{Cobs}$ and $\Gamma_{Sobs}$, respectively. As they use the same meteorological fields, the difference between $\Gamma_{bir}$ and $\Gamma_{trj}$ should correspond to the advantages and limitations of each estimation method. We begin our discussion with the comparison between these two.

The vertical distributions of mean age are illustrated in Fig. 13, comparing $\Gamma_{bir}$ (green triangles) and $\Gamma_{trj}$ (black crosses) with $\Gamma_{Cobs}$ (blue circles) and $\Gamma_{Sobs}$ (magenta squares) given by S18. Generally, $\Gamma_{bir}$ tends to be older than $\Gamma_{trj}$, but they are in good agreement below 26 km; $\Gamma_{bir}$ is only 0.2 years older than $\Gamma_{trj}$ from 18 km to 26 km. Above 26 km, however, $\Gamma_{bir}$ is $\sim 1.2$ years older than $\Gamma_{trj}$. Such a difference could result from the following factors. One is related to the height of the model top. In the BIR method, the transport calculations extend from the surface to approximately 90 km, while the trajectory

calculations are restricted to the region below 1 hPa ($\sim 50$ km). The significance of the different calculation domains depends on the proportion of air parcels that reach the top boundary during the trajectory calculations. In the case of Fig. 9, for example, 252 out of 8559 parcels have reached the top boundary without crossing $Tr_{top}$ during the 10-year backward calculation. They descended from the mesosphere (above 1 hPa) to the high-latitude stratosphere along the deep branch of the BDC, rapidly migrated to the tropics by quasi-isentropic motion, and reached the middle stratosphere following slow diabatic ascent in the

tropical stratosphere. The outcome from the omission of this pathway from the age estimation is qualitatively consistent with the discrepancy relative to $\Gamma_{Sobs}$ to be discussed later.

The second factor is the contribution of the long tail. In the BIR method, more than 50 % of the mean age comes from the tail (Li et al., 2012b). On the other hand, the tail correction is not critically important in dealing with the tropical stratosphere with 10-year back trajectories (Sect. 3.2). Such a difference may be explained by the fact that the trajectories are driven solely

by resolved wind while diffusion and unresolved mixing parameterized in the ACTM incorporate the entrainment of "ancient" air leading to the increase of mean age in the BIR method.

As expected from the good agreement of the trajectory-derived $CO_2$ profile with observations (Fig. 10(a)), $\Gamma_{Cobs}$ is reasonably well reproduced by $\Gamma_{trj}$. That is, both $\Gamma_{Cobs}$ and $\Gamma_{trj}$ increase gradually up to 25 km and stay nearly constant above it. On the other hand, $\Gamma_{bir}$ continues to grow up to 29 km, deviating from $\Gamma_{Cobs}$ at 28 and 29 km. $\Gamma_{Sobs}$ also increases almost linearly

up to 24 km as in $\Gamma_{Cobs}$. One of the reasons for the difference between $\Gamma_{Cobs}$ and $\Gamma_{Sobs}$ may be the strong seasonal variation in tropospheric $CO_2$. As the lag-method cannot be applied to such periodic tracers (Waugh and Hall, 2002), S18 used hypothetical age spectra defined by the inverse-Gaussian distribution parameterized as a function of $\Gamma$ and $\Delta$ (Waugh and Hall, 2002). $\Gamma_{Cobs}$ was determined by choosing the value that best approximates the observed $CO_2$ mole fraction by convolutions of the observed tropospheric reference record and the hypothetical age spectra. As the age spectra have two unknown parameters while only





one known value ($CO_2$ mole fraction) is available, they assumed the relationship $\Delta^2/\Gamma = 0.7$ years as a constraint to solve the problem. The value of 0.7 years, suggested by Hall and Plumb (1994) from the results of a stratospheric GCM, is found to give the best overall agreement between $SF_6$-derived and $CO_2$-derived mean ages in the northern mid- and high latitude stratosphere where the $CO_2$-mean age ranges from two to five years (Engel et al., 2002). As is shown in Table 2, however,

the $\Delta^2/\Gamma$-ratio estimated from our age spectra is altitude dependent and generally smaller than 0.7 years. A series of $\Gamma_{Cobs}$ calculations is conducted as sensitivity tests by sweeping the $\Delta^2/\Gamma$-ratio in the range from 0.05 to 0.70 years. Fortunately, the fluctuations of $\Gamma_{Cobs}$ stay within the observational uncertainties in the parameter range obtained from the BIR and trajectory methods and there is scarcely any need to revise the former estimates published in S18. However, there is an important finding in that $\Gamma_{Cobs}$ becomes a multi-valued function of $CO_2$ mole fraction at some specific altitude and time due to the propagation of

the seasonal cycle of $CO_2$. Such an occurrence is not unusual in the tropical lower stratosphere where the tropospheric seasonal cycle is robust to mixing. In the case of the CUBE/Biak samples, for example, the $CO_2$ mole fraction of $\sim 397.9$ ppm falls into this singularity if the $\Delta^2/\Gamma$-ratio is as small as 0.05 years. In such cases, $\Gamma_{Cobs}$ cannot be uniquely determined by the observed $CO_2$ mole fraction, emphasizing the importance of the use of realistic value of the $\Delta^2/\Gamma$-ratio in the age estimation from $CO_2$. In this context, it is worth emphasizing that the age estimation from clock tracers is influenced not only by the value of the

$\Delta^2/\Gamma$-ratio but also by the fine structure of age spectra that may deviate from the inverse-Gaussian distribution. Therefore, the uncertainties of age estimation based on the clock tracer observations should be considered carefully in terms of the reliability of age spectra assumed in those methods. Fine structures of age spectra and their implications for the $\Delta^2/\Gamma$-ratio are discussed by comparing with the corresponding trajectories in the Appendix.

Above 25 km, $\Gamma_{Sobs}$ is much larger than $\Gamma_{Cobs}$, although near constant values are commonly observed as in $\Gamma_{trj}$. On the other

hand, $\Gamma_{bir}$ increases almost linearly with altitude up to 28 km. The linear increase of $\Gamma_{bir}$ may suggest that the diffusivity of our ACTM is a little too high in the tropical stratosphere, although it is good enough to reproduce the overall features. While the age estimation from $CO_2$ samples may partly be compromised by the strong seasonal variation of $CO_2$ in the troposphere, the systematic difference between $\Gamma_{Cobs}$ and $\Gamma_{Sobs}$ in the upper stratosphere is often attributed to the mesospheric loss of $SF_6$ (Andrews et al., 2001a; Waugh and Hall, 2002; Stiller et al., 2012, S18). That is, the downward transport of $SF_6$-depleted

mesospheric air is misinterpreted as the persistence of the aged air that entered the stratosphere when the tropospheric $SF_6$ mole fraction was low. The higher mole fractions (Fig. 10) and younger age (Fig. 13) of $SF_6$ derived by back trajectories against those derived from observations (S18) are consistent with this idea since the air parcels of mesospheric origin are missing in the estimation from trajectory calculations.

## 5   Summary

An atmospheric chemistry transport model (ACTM) has been used, together with ERA-Interim reanalysis data, to investigate the mean age profiles estimated from the $CO_2$ and $SF_6$ concentrations observed during the CUBE/Biak field campaign. The age spectra are estimated by two methods, the boundary impulse response (BIR) method and Lagrangian backward trajectory calculation, relying on the meteorological fields simulated by the ACTM. The BIR method has an advantage in estimating the



age spectra in that the simulated transport processes are inclusive of unresolved mixing and diffusion. The Lagrangian method is capable of estimating the vertical profiles of $CO_2$ and $SF_6$ concentrations and the water vapor "tape recorder" in addition to that of the mean age, distinguishing the contribution of air particles that have taken specific paths before reaching the point of interest. The reproducibility of the observed profiles of $CO_2$, $SF_6$ and the "tape recorder" confirms the reliability of the

transport calculations. The application of these two methods that rely on the same transport field has proved useful in assessing the mean age profiles derived from clock tracer concentrations.

The important findings are summarized as follows. The profile of $CO_2$-mean age, $\Gamma_{Cobs}$, is reproduced reasonably well by back trajectories, $\Gamma_{trj}$. The reproducibility of the BIR method, $\Gamma_{bir}$, is limited to below 25 km. At 22 km, $SF_6$-mean age, $\Gamma_{Sobs}$, is approximated better than $\Gamma_{Cobs}$ by both $\Gamma_{bir}$ and $\Gamma_{trj}$, suggesting that the elimination of the seasonal variation of the

tropospheric $CO_2$ may be incomplete in the tropical lower stratosphere. As for $SF_6$, the observed concentrations and $\Gamma_{Sobs}$ are reproduced only below 24 km by $\Gamma_{trj}$. Above 25 km, observed concentrations are much smaller and $\Gamma_{Sobs}$ appears much larger than those from Lagrangian estimation. These results are consistent with the notion that $\Gamma_{Sobs}$ derived from stratospheric air samples is overestimated by the mixing in of $SF_6$-depleted mesospheric air, since those air parcels that are traceable to the mesosphere without passing through the troposphere are missing in the Lagrangian age estimation. The values of $\Gamma_{bir}$, closer

to $\Gamma_{Sobs}$ but much larger than $\Gamma_{Cobs}$ and $\Gamma_{trj}$, at 28 to 29 km may be interpreted in terms of the diffusivity of the transport field (effective only for BIR calculations) being a little too high in the upper tropical stratosphere.

This study urges caution in the use of long-duration kinematic trajectories in age estimation. The application of nudging one-hour averaged wind and temperature fields at one-hour intervals reproduces the observed features reasonably well, while the use of instantaneous six-hour interval data failed to do so. This study reconfirms the difficulty in estimating the mean age

from clock tracer observations. The widely used relationship between the spectral width $\Delta$ and the mean age $\Gamma$, $\Delta^2/\Gamma = 0.7$ years, is not supported in our ACTM, suggesting that this ratio is model dependent. Observations of multiple clock tracers and other variables independent of mean age, such as the gravitational separation, will be useful to constrain the stratospheric transport features.

## Appendix A: Fine structure of age spectrum reflecting the pathway difference

The assumption that the $\Delta^2/\Gamma$-ratio is a constant indicates that the spectral width measured by $\Delta$ is solely determined by the mean age $\Gamma$ with spectral broadening occurring with increasing $\Gamma$. However, this is not always the case in our experiments as can be seen from Table 2. We will take a close look at the two lowest altitudes for which the $\Delta^2/\Gamma$-ratios derived from trajectories are quite small. The tail correction is negligible and is ignored here. BIR spectra are not considered since fine structures unresolvable on a monthly mean basis are discussed.

Figure A1 shows the age spectra (left panels) and some examples of back trajectories (right panels). $\Gamma$ increases from 36 days for the lowest sample (Sample 1) to 47 days for the second lowest one (Sample 3). However, $\Delta$ is much smaller in Sample 3 than in Sample 1, reflecting the small amount of old air. The contrast between the two samples corresponds to the difference in the trajectories (right panels). In the case of Sample 1 (top panel), air parcels can leak from the tropical "pipe"



(in a backward sense of time) to midlatitude (shown in blue and green) and even to high latitudes (in red) at the altitude of initialization, although the large majority (dark colours) stay within $\pm 30°$ of the equator. It is interesting to see the leakage predominantly occurs towards the SH, which suggests that seasonal variation is important for the structure of TTL circulation. The air parcels of Sample 3 (bottom), on the other hand, need to descend (again in a backward sense) slowly to the TTL

where the pathways to midlatitudes open. It is rather surprising that the age spectrum has a bimodal structure even though the $\Delta^2/\Gamma$-ratio is as small as 0.08 years. The cause of this modality is explored by colour-coding the age spectrum and trajectories corresponding to the first and second peaks, and the tail of the spectrum, in black, red, and green, respectively. Following the residual stream function upstream (Fig. 1), air parcels are deflected slightly to the north during their descent to the TTL. The portion of parcels constituting the first spectral peak reaches $Tr_{top}$ without migrating to midlatitudes. The second peak is

composed of those particles that have leaked from the tropical "pipe." Depending on the seasonal variation of the circulation pattern and the latitude of descent, the predominance of the leakage to the south seen in Sample 1 disappears. Those particles that have escaped from the tropical "pipe" at higher altitudes travel longer in the stratosphere to make up the tail of the age spectrum.

The fine structure of the age spectrum thus reflects the transport features in the TTL and the lower stratosphere. As mixing

and diffusion, in addition to the model-resolved wind that has driven trajectories in the above calculations, contribute the tracer transport, many of the details may be obscured in the real atmosphere. Despite such limitations, however, these results are enough to emphasize the importance of for the search for an accurate age spectrum.

*Author contributions.*  HTN and FH designed the experiments, prepared the manuscript and illustrations. KI performed the simulations and revised the manuscript. SS carefully revised the manuscript.

*Competing interests.*  The authors declare that they have no conflict of interest.

*Acknowledgements.*  HTN wishes to express her gratitude to members of her laboratory, especially D. Belikov, X. T. Nguyen Vinh, and S. Mimura for their encouragement and discussion during her stay in Hokkaido University. KI and FH are grateful to K. Miyazaki for consultation on designing the BIR simulations on ACTM. Discussion with S. Ishidoya greatly inspired our study. This work was supported by the Japan Society for the Promotion of Science, Grant-in-Aid for Scientific Research (S) 26220101.



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



**Figure 1.** Monthly mean distributions of surface-released pulse tracers (ppbv) in February (top) and August (bottom) corresponding to (a, c) the 2nd and (b, d) the 14th month from release in January and July, respectively. The dotted yellow line is the tropopause position (WMO-defined). Area covered by the purple lines represents the TTL. White curves show the TEM residual stream function (kg m$^{-1}$ s$^{-1}$).



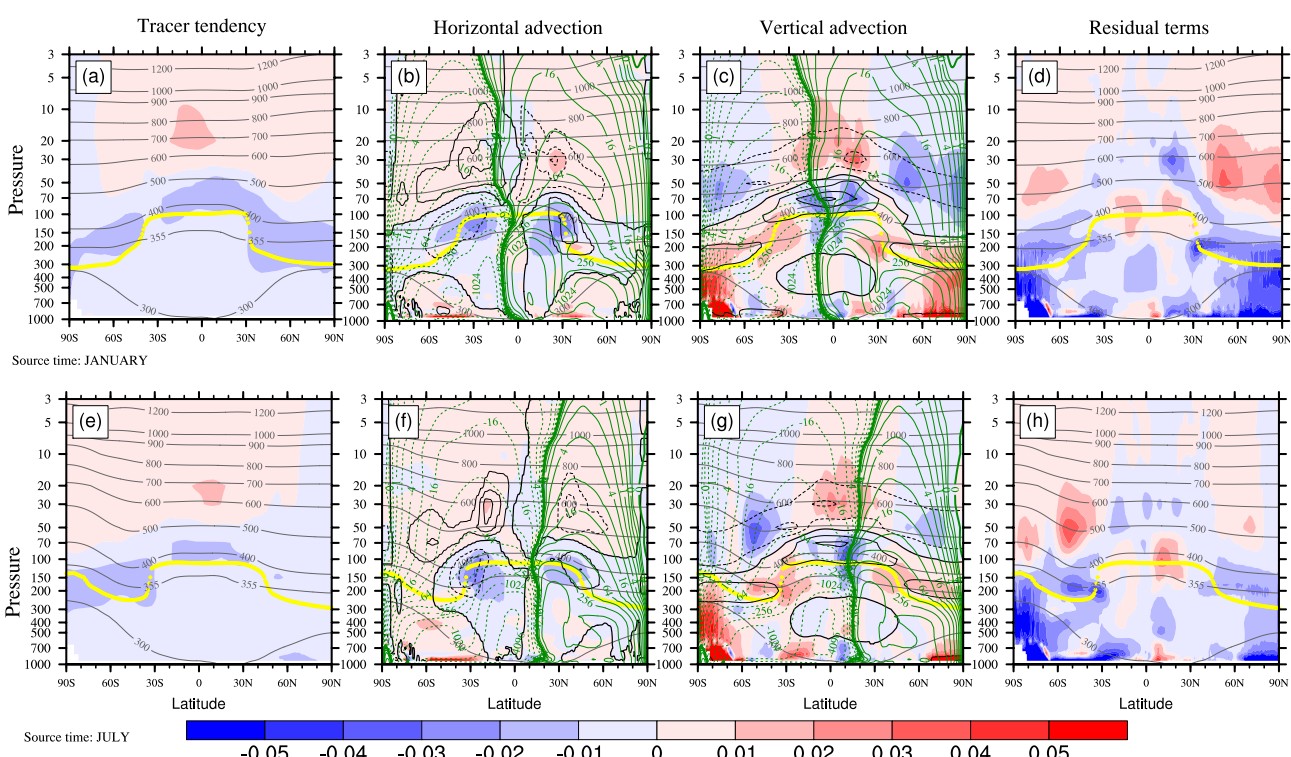

**Figure 2.** Meridional distributions of the averaged tendency for the one-month period from the 14th to 15th month after release in January (top) and July (bottom) ((a) February to March, (e) August to September). Other panels are the decomposition into (b, f) the horizontal and (c, g) the vertical advection, and (d, h) the remaining terms (ppbv month$^{-1}$). Grey lines are potential temperature (K). Black lines in (b, f) and (c, g) are the horizontal and vertical tracer gradients, respectively (solid for positive and dashed for negative values). The contour interval is $1 \times 10^{-8}$ ppbv m$^{-1}$ for horizontal and $1 \times 10^{-5}$ ppbv m$^{-1}$ for vertical gradient. Zero lines are omitted. The yellow line is the position of the thermal tropopause. Green curves represent the TEM residual stream function (kg m$^{-1}$ s$^{-1}$).

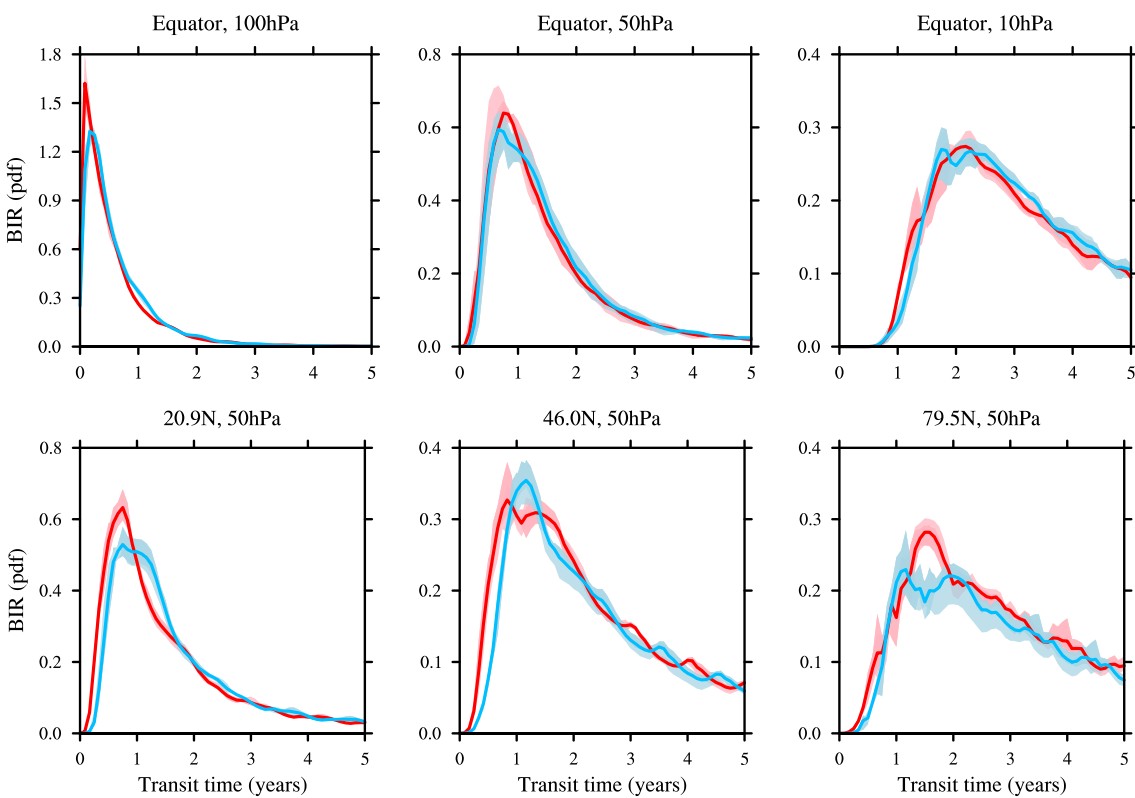

**Figure 3.** Evolution of January (red) and July (blue) BIRs on 100, 50, and 10 hPa over the equator (top row), and on 50 hPa at 20.9, 46.0 and 79.5° N (bottom row). The thick lines are the average of five BIRs for tracers released from 2005 to 2009, while the shading shows the BIR range among the five releases.

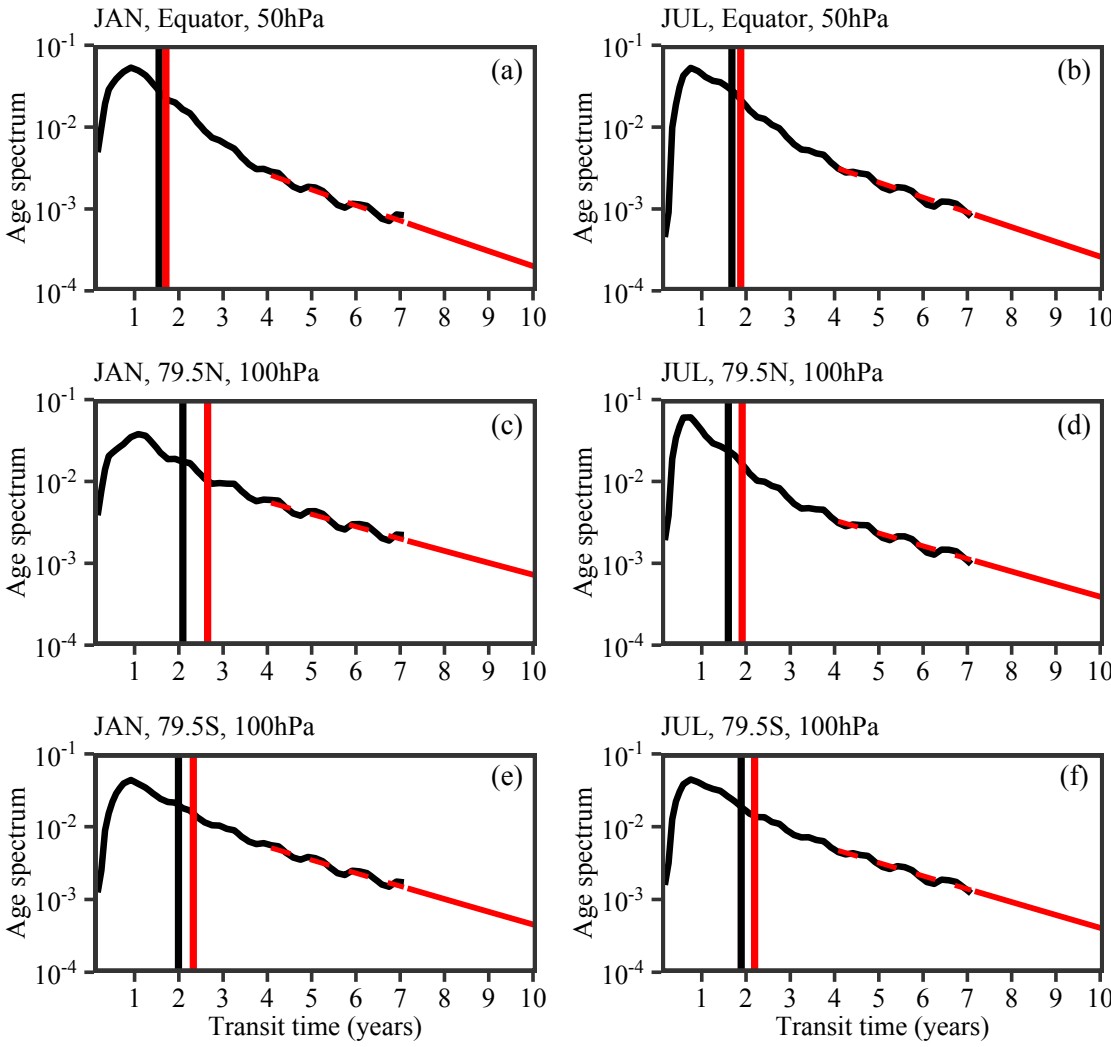

**Figure 4.** Multi-year averaged age spectrum (year$^{-1}$) at (top) 50 hPa over the equator, and at 100 hPa (middle) 79.5° N, and (bottom) 79.5° S in January (left) and July (right). The black lines show the age spectra and the red lines are the extrapolation for tail correction (see text for details). The vertical lines show the mean age with (red) and without (black) tail correction.




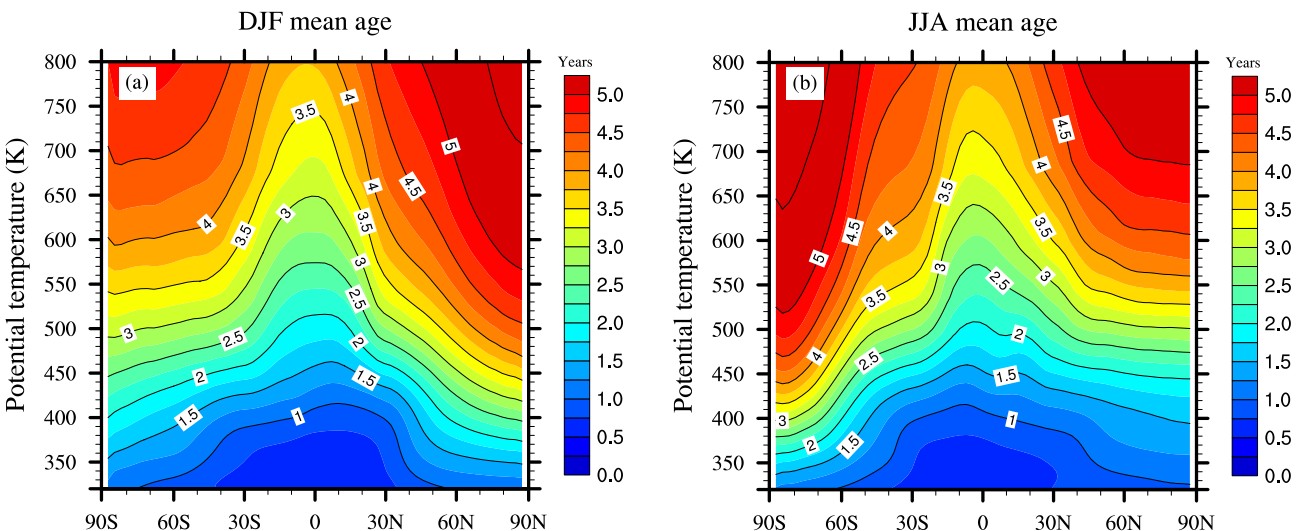

**Figure 5.** Zonal mean distribution of three-year averaged mean age in NH winter (DJF) and summer (JJA). The potential temperature (K) is taken as the vertical coordinate.



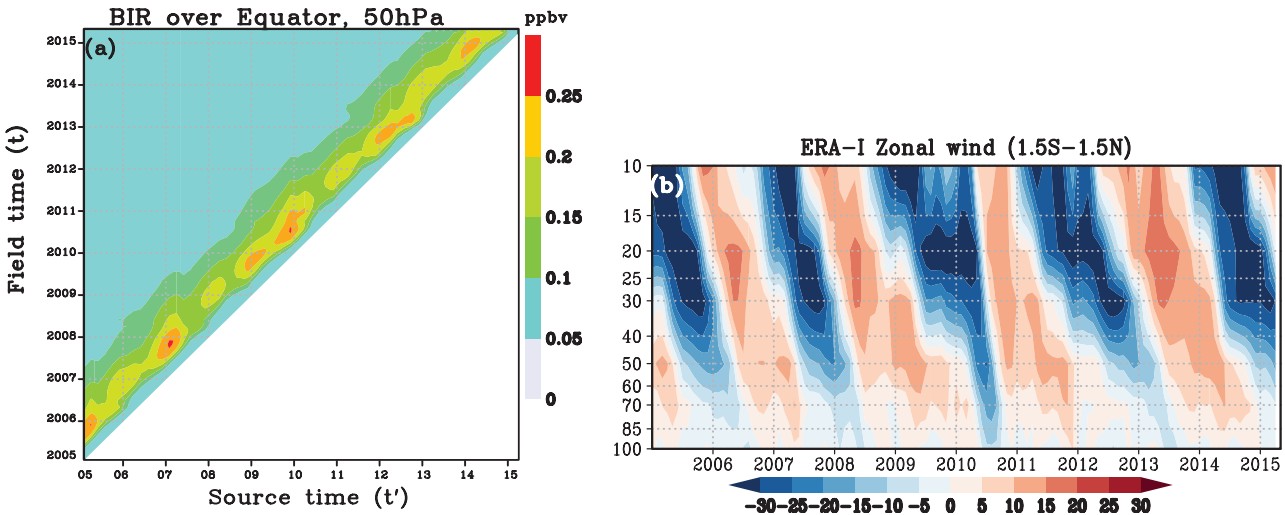

**Figure 6.** (a) BIR map at 50 hPa over the equator constructed using tracers released from January 2005 to March 2015 in the source regions at the tropical surface (15° S–15° N). (b) A time–height section of mean zonal wind (m s$^{-1}$) over the equator.

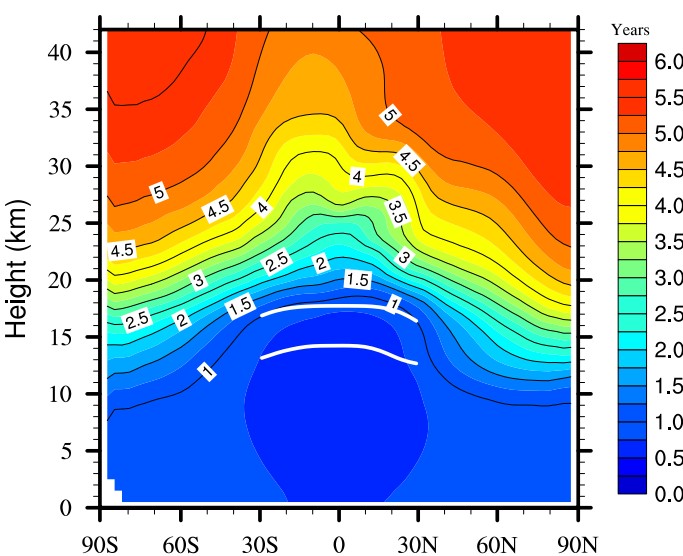

**Figure 7.** Latitude–height section of the mean age ($\Gamma_{corr}$) in March 2015 estimated using tracers released from January 2005 to March 2015 at the tropical surface (15° S–15° N). The age correction is applied using the transit time from the 4th to 10th years. The 355 K and 400 K isentropes are shown as white lines bounded between 30° N and S indicating the location of $Tr_{top}$ and the top of the TTL, respectively.

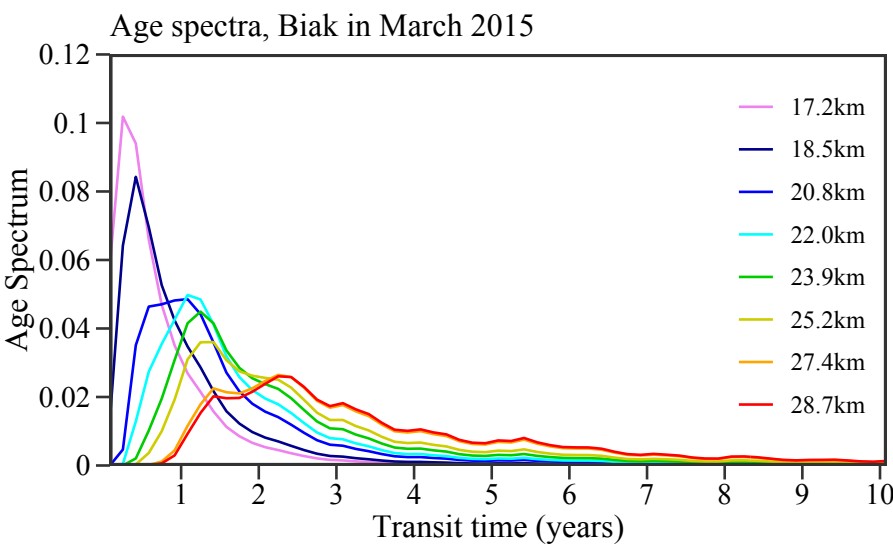

**Figure 8.** Age spectra (pdf) corresponding to the altitudes of eight cryogenic air samples acquired during CUBE/Biak 2015.



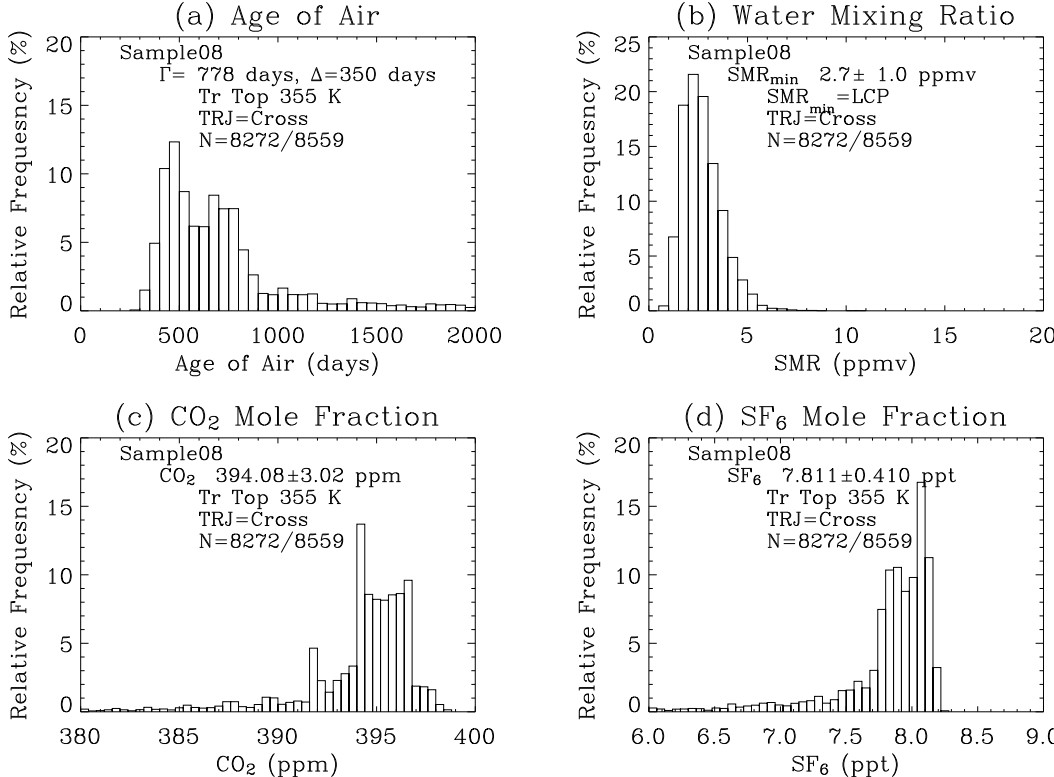

**Figure 9.** Spectra of (a) age of air (days) and (b) water mixing ratio (ppmv), and mole fractions of (c) $CO_2$ (ppm) and (d) $SF_6$ (ppt) estimated by trajectory calculations corresponding to Sample 8 from CUBE/Biak observations. The mean age $\Gamma$ and the width of the age spectrum $\Delta$ (Hall and Plumb, 1994) are 778 days and 350 days, respectively. The top of the troposphere ($Tr_{top}$) is taken to be the 355 K isentrope and the saturation mixing ratio at the Lagrangian cold point is used to determine the final water amount carried to the stratosphere. Kinematic backward trajectories for 3652 days (10 years) relying on ACTM simulations nudged to ERA-Interim meteorological fields are used.



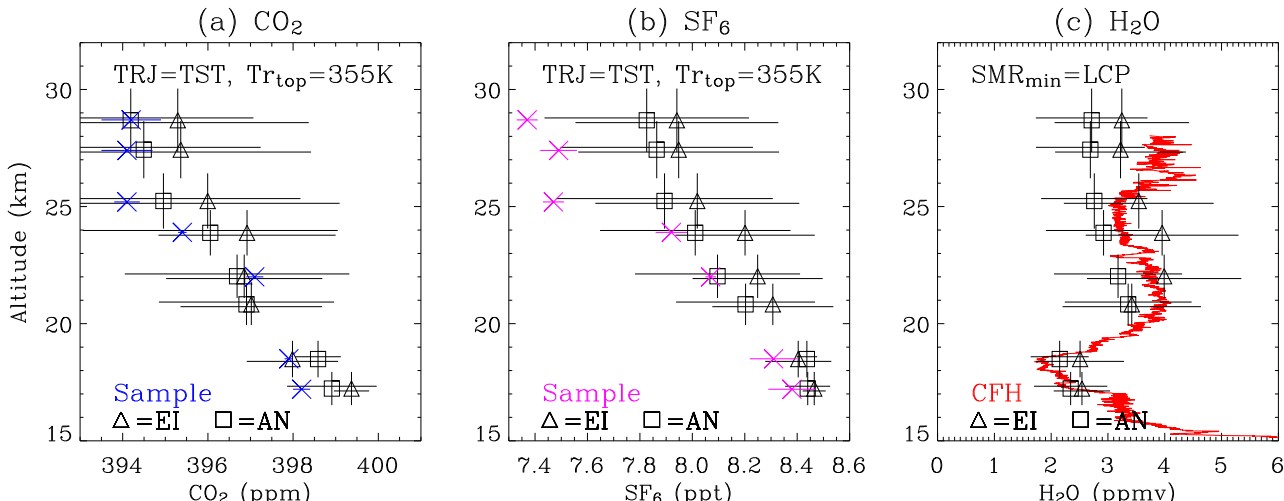

**Figure 10.** Vertical profiles of mole fractions of (a) $CO_2$ (ppm), (b) $SF_6$ (ppt), and (c) water vapor mixing ratio (ppmv). Observational estimates from air samples are shown in colour (S18), while the red line shows the observed water vapor profile depicted in H18. Triangles and squares denote the estimates from EI and AN trajectory calculations, respectively. The horizontal bars are the intervals of uncertainty expressed as $\Delta$ (Hall and Plumb, 1994; Waugh and Hall, 2002) drawn by shifting 0.1 km downward and upward for EI and AN, respectively, for visual clarity, while the vertical bars are the ranges for air sampling and the initialization height range for trajectory calculations.

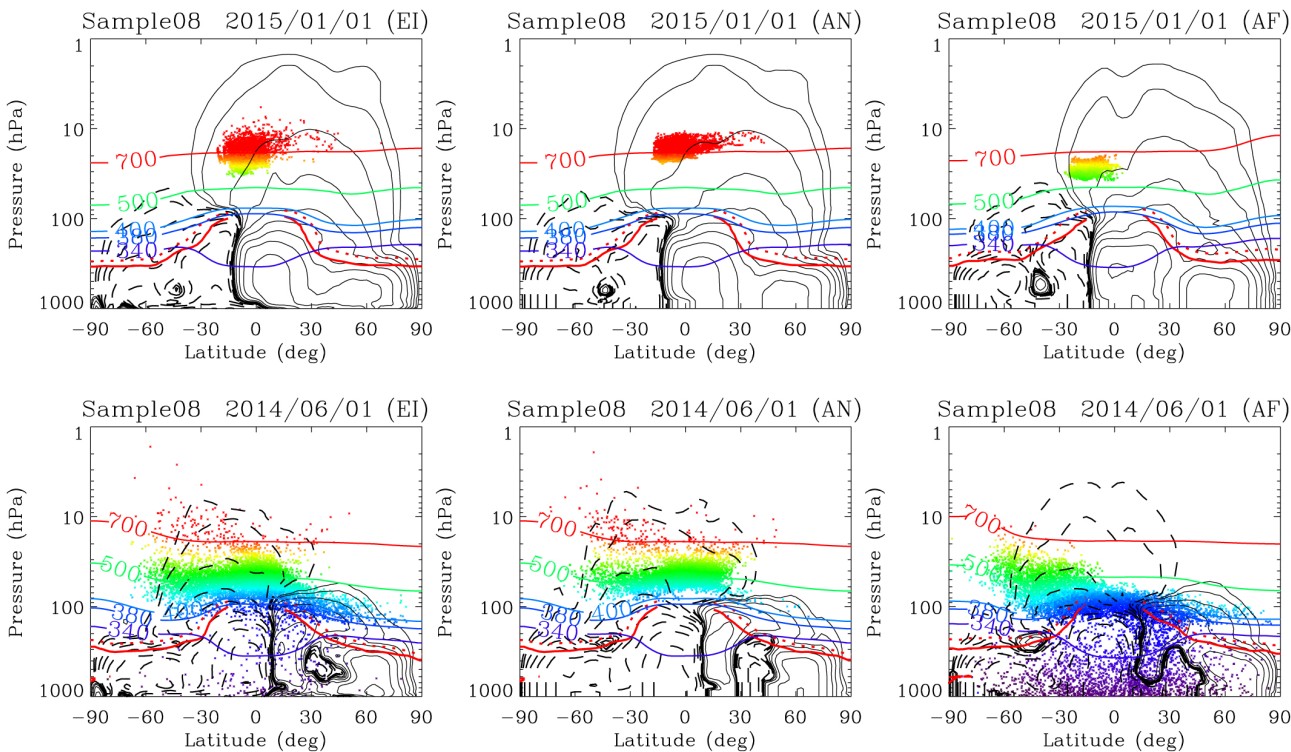

**Figure 11.** Snapshots of the meridional location of air parcels corresponding to Sample 8 on (top) 1 January 2015 and (bottom) 1 June 2014 calculated by backward trajectories superposed on the residual stream function (black contours at $\pm(0.1, 0.2, 0.5, 1, 2, 4, 8, 15, 30, 50)$ kg m$^{-1}$ s$^{-1}$; solid and dashed contours are positive and negative values, respectively.) The air parcels and the contours of zonal mean potential temperature are colour-coded. The thick red lines are the zonal mean potential vorticity ($2 \times 10^{-6}$ m$^2$ s$^{-1}$ K kg$^{-1}$ (PVU) in solid and 4 PVU in dashed lines) as proxies of the extratropical tropopause drawn outside of $15°$ latitude.



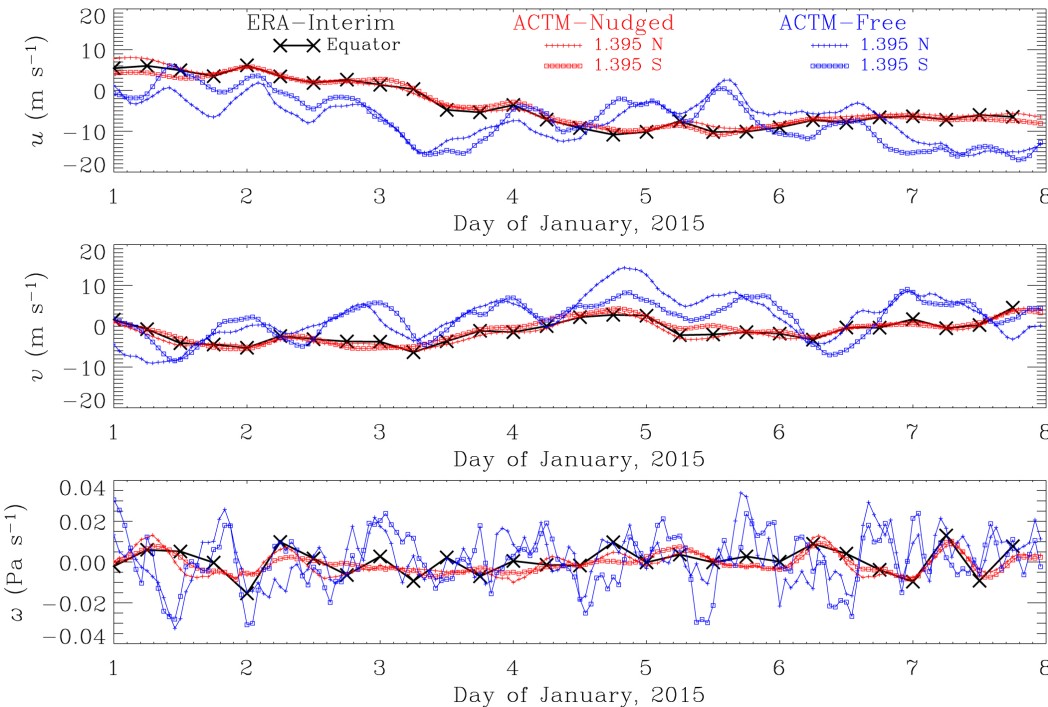

**Figure 12.** Time series of the zonal ($u$), meridional ($v$), and vertical ($\omega$) wind components at grid points $0°$ longitude near the equator from ERA-Interim reanalysis (black crosses), together with those from ACTM nudged to ERA Interim (red) and the ACTM free run (blue) at $1.395°$ N and S (the latitudes closest to the equator from the available Gaussian latitudes). ERA-Interim winds are provided as instantaneous values every 6 hours, while ACTM winds are given as hourly averages.



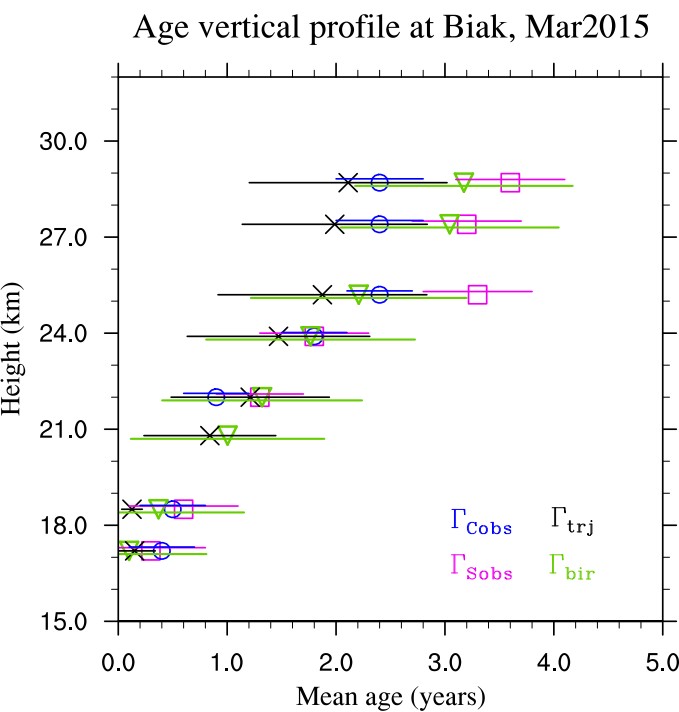

**Figure 13.** Comparison of the vertical profiles of mean age estimated by the BIR method ($\Gamma_{bir}$; green), back trajectories ($\Gamma_{trj}$; black), and cryogenic samples of $CO_2$ ($\Gamma_{Cobs}$; blue) and $SF_6$ ($\Gamma_{Sobs}$; magenta) published in S18. Horizontal bars are the uncertainties plotted by shifting 0.1 km downward and upward for $\Gamma_{bir}$ and $\Gamma_{Sobs}$, respectively, and shifting 0.12 km upward for $\Gamma_{Cobs}$, for visual clarity.

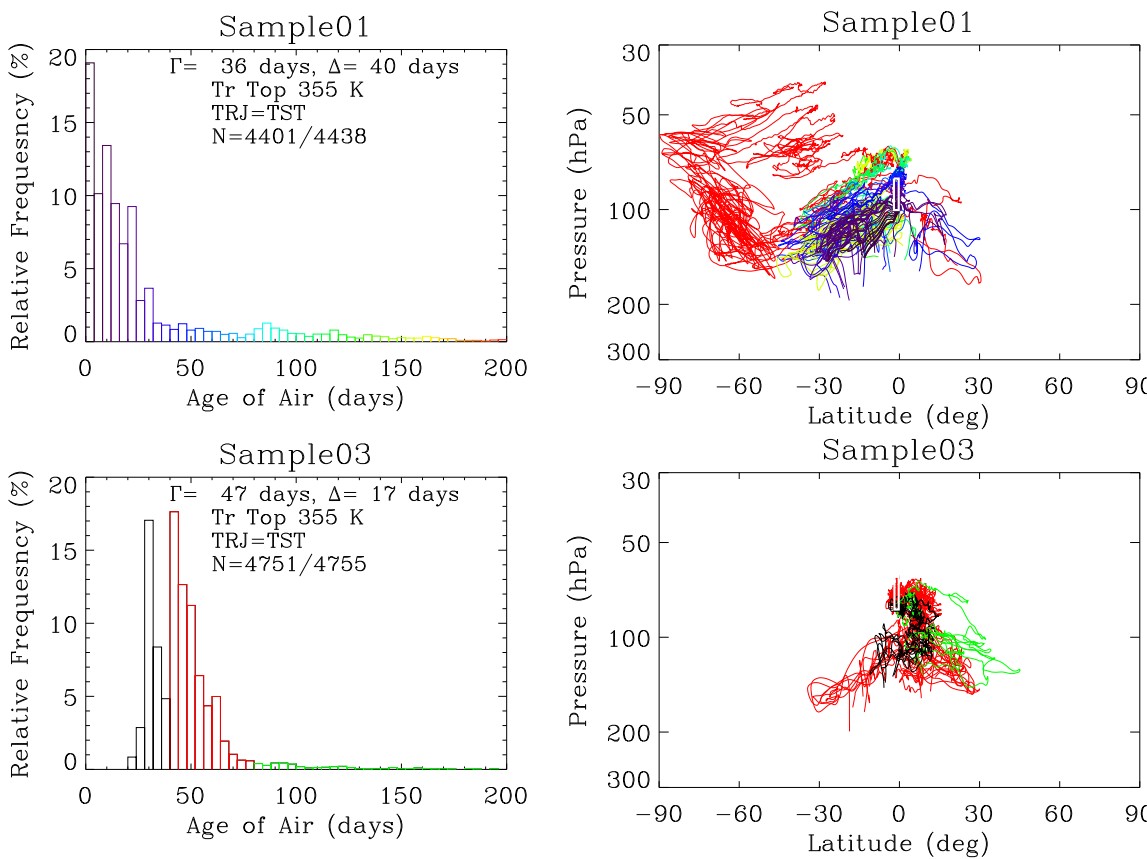

**Figure A1.** Age spectra (left) and meridional projection of AN back trajectories drawn from the initial position (located in the white rectangle) to the first crossing of $Tr_{top}$ (right) colour-coded by the age of air shown in the left panel of the same row. Upper panels are for the sample at the lowest altitude (16.5–17.9 km) and the bottom panels are for the second-lowest (17.1–19.3 km). The colour in the bottom panel is simplified to three scales corresponding to less than 40 days (black), between 40 and 80 days (red), and greater than 80 days (green). Those illustrated are the TST trajectories limited to 1 % (top) and 0.5 % (bottom) of the total population so as to avoid excessive overlap in the meridional projection.



**Table 1.** Experimental conditions for backward trajectory calculations. H18 refers to those of Hasebe et al. (2018), while EI, AN, and AF are those of the present analysis corresponding to ERA-Interim, ACTM nudged to ERA-Interim analysis, and ACTM free run, respectively.

| Run | H18 | EI | AN | AF |
|---|---|---|---|---|
| Meteorological Field | ERA-Interim pressure-level data | | nudged to ERA-Interim | 10-year free run initialized by ERA-Interim |
| Pressure Levels (hPa) | 1000, 975, 950, 925, 900, 875, 850, 825, 800, 775, 750, 700, 650, 600, 550, 500, 450, 400, 350, 300, 250, 225, 200, 175, 150, 125, 100, 70, 50, 30, 20, 10, 7, 5, 3, 2, 1 | | 140, 130, 120, 115, 110, 105, 95, 90, 85, 80, 75 in addition to the left | |
| Horizontal Resolution | $1.5^\circ$ longitude $\times 1.5^\circ$ latitude | | $2.8125^\circ$ longitude $\times$ 64 Gaussian latitudes | |
| Time Resolution | 6 hours instantaneous | | 1 hour average for an hour | |
| Integration | 1200 days | 3652 days | | |
| Initialized Particles | 1134 (sample 1) to 2187 (sample 8) | 4438 (sample 1) to 8559 (sample 8) | | |

**Table 2.** Values of $\Delta^2/\Gamma$ corresponding to the air samples estimated from the age spectra. Note that $\Gamma$ used for this estimation is not $\Gamma_{bir}$ (Fig. 13) but $\Gamma_{corr}$ (Sect. 3.1).

| Altitude (km) | BIR method (year) | Back trajectories (AN) (year) |
|---|---|---|
| 17.2 | 0.65 | 0.25 |
| 18.5 | 0.59 | 0.08 |
| 20.8 | 0.47 | 0.42 |
| 22.0 | 0.42 | 0.43 |
| 23.9 | 0.38 | 0.47 |
| 25.2 | 0.34 | 0.49 |
| 27.4 | 0.27 | 0.37 |
| 28.7 | 0.26 | 0.40 |