# Peer review of "Transport model diagnosis of the mean age of air derived from stratospheric samples in the tropics"

_Atmospheric Chemistry and Physics, 2020_

## Referee Comment (RC1) · Anonymous Referee #1 · 2 Jun 2020

**Review on:**
**"Transport model diagnosis of the mean age of air derived from strato-spheric samples in the tropics", ACPD, 2020, by Nguyen, H. T. et al..**

In their paper, Nguyen et al. apply two methods (BIR and backward trajectory calculation) to model age of air (spectra) as well as mole fractions of $CO_2$, $SF_6$ and water vapour. In the paper, first the model results are evaluated and subsequently some results are compared with data from a measurement campaign. The comparison works reasonably well, with some discrepancies that mostly can be explained.

[Figure]

The overall idea of the paper is good and the method is elaborate. Some of the results are also interesting and can contribute to foster science in this field, although it is for example expectable that chemical $SF_6$ depletion will not allow direct comparison in the upper stratosphere if it is not included in the model. It is good to carve out which of the modelling methods is suitable to tackle which science question.

However, the paper is chaotic and does not provide the necessary information to follow. Almost nothing is reported about the measurements of that campaign and what are the points that were supposed to be investigated with them. The model description is unclear, I do not understand why sometimes nudging is described, while the authors apply a CTM, which usually is driven (not nudged) by reanalysis data. That is very confusing. The evaluation of the results is pretty lengthy and should be reduced to about two figures. If need be, the rest can be banished to a supplement (maybe together with the appendix). I like the idea of explaining measurements with modelled AoA spectra, but at the end, that is only a minor part in the paper and is only partly successful (partly due to the sinks). However, my main point really is that the study does not follow a clear research question. The reader can be lost due to that. What is it exactly that is puzzling you about the measurements? Why do you think the applied method can help to answer that question and how do you plan on pursuing that? How can additional information about transport processes be gained through that? What is your contribution to improve the undertanding of the underlying processes at the end and how does that fit into existing literature? Some of this information information is lacking, some is spread somewhere across the manuscript, and the reader has to put the pieces together. Further, so many different points are applied, AoA, its spectra, the $CO_2$ and $SF_6$ mole fractions as well as water vapour and the cold point are thrown together, but it does not clearly shine through that all these measures are needed to make the conclusions that are drawn at the end. Also, different models, free running, nudged simulations and/or the CTM and the two diagnostic methods, all that leads to confusion and does not help to get to the point. Additionally, intense use of non-intuitive abbreviations complicate reading and also in the results section, many points

are thrown together and a clear focus is missing.

Hence, I would suggest the authors to completely revise the manuscript and then submit it again. I think the study can help to advance our understanding of stratospheric transport and the methods that can be used to investigate it if it is presented and structured properly. Please start with one or more clear research questions that can be answered with this method and build everything around that. Use only the methods needed, describe them clearly, and then take the reader point by point towards the conclusion. Please also consider my additional comments that I am making below.

**Additional comments:**

- P1L2 A CTM is not nudged! A CTM uses some meteorological fields for describing transport. This is totally confusing and it is not clear to me what is actually done in this study, because later also you talk about GCM and CTM. Please clarify what that is and what you do throughout the paper.

- P1L4: Change "a single" to "the chemistry transport model"

- P1L3-5 The sentence is unclear. Are there discrepancies between the two models or between models and observations? And the following sentence starts with a "This", but it is unclear what the "this" refers to, to the usefulness, or to the discrepancies.

- P1L7: But where is the connection between the water vapour tape recorder and the mean age here?

- P1L8: Change "the reality" to "good quality" or something alike

- P2L12: Please consider also the newer publication by Engel et al. from 2017 (10.5194/acp-17-6825-2017)

- P2L18: observations and models

- P2L18: With "sampling of clock tracers" do you mean $SF_6$ and $CO_2$? Or more tracers? Can you elaborate a little more on the campaign, please, like what, how, how long...?

- P2L21: What exactly is it that is puzzling you about these measurements? This should be central in all parts of the paper.

- P2L25: But that is why the Green's function is used to flatten out these non-linearities. Please see and possibly mention Fristch et al. 2020 (10.5194/acp-2019-974) and citations therein.

- P3L17-18: So is it a CTM or a GCM now?

- P3L23: It would be easier to use the name of the model from here on, instead of "ACTM", if the model has a name.

- P3L27: How did the model perform in that inter-comparison? Was it somewhere around the multi-model mean or was it an outlier?

- P4L5: "several years". Please be precise, for the sake of reproducibility. Did you use ERA-I data of year 2004 for that and repeat that year for ? years?

- Sects. 2.3 and 2.4 are just the evaluation of the model. Firstly, this should be reflected in the section headers, and secondly, these sections should be considerably shortened and/or moved partly to the supplement. I suggest to reduce the number of figures from 5 to 2.

- These abbreviations, particularly "AF" and "AN" do not make much sense to me.

- Fig. 2 I do not fancy that the streamfunction is shown again here, it was shown in Fig. 1 already and does not help much, instead, it disturbs the view on the tendencies. I suggest to remove it.

- P5L15 What do you mean by "selected tracers"?

- P7L1 ...temperature move upward over time.

- P7L6 backward

- P7L15-17 Unclear and awkward phrasing, please rephrase. Plus, what is meant by "not simple"?

- P7L25-26 Remove parentheses around dates

- P7L28 change "drives the tracers upward" to "intensifies the upward tracer transport"

- P7L31 "The vertical axis is ...." What is that sentence supposed to mean? It makes no sense to me.

- P7L32 What is a latitudinal split? Please explain clearly what you talk about. Moreover, why is that important now, you explained the QBO topic already before.

- P8L5 ...the spread of the transit times ($\Delta$)...

- P8L9 A bundle! Please be more quantitative.

- P8L9 the spectra of AoA, $CO_2$ and $SF_6$ mole fractions? You are mixing up something here, please be specific.

- P8L11 Can you still give a very brief description of the method of analysis please.

- P8L12-14 I do not understand what this sentence is supposed to mean. Please rephrase it and sharpen the message.

- P8L14 Where do these additional levels come from?

- P8L21 What is CONTRAIL data? Please describe!

- Fig.9 Water "vapour"! Or is ice included too? (Throughout the paper!)

- P9L2 and L5 You already defined these abbreviations above.

- P9L3 But what were the problems in H18? Can you provide a quick introduction? Without that, it is almost impossible to follow.

- P9L18 "It is interesting". Does that mean the other results are not interesting?

- P9L18-32 Using these abbreviations this way make it almost impossible to follow.

- P9L25 How is that question linked with the general idea of the paper?

- P10L5-9 Even more abbreviations that totally disturb readability.

---

## Referee Comment (RC2) · Anonymous Referee #2 · 3 Jun 2020

The paper by Nguyen et al. investigates different methods with which observations of mean age tracers and mean age of air (AoA) can be reconstructed. While the study is certainly in the scope of ACP and the subject is of significant scientific interest, there are a number of problems I see in the investigations presented here. In my view, the authors misinterpret some aspects of AoA, as detailed below. Further, in my view some aspects are presented (e.g. related to Figures 1 and 2) which are not taken up in the discussion or conclusions of the manuscript and are not necessary for the understanding. Other aspects like experimental details (e.g. where, when and how were the samples taken) and how was AoA calculated from the observations is omitted. There are also a number of important recent papers, which are not included in

the discussion (see details given below). Due to these issues, I believe that the paper is not ready for publication, but needs major revisions before it can be considered for publication.

**Major comments**

*Clock tracers and derivation of AoA*

I believe that there is a misinterpretation on what is commonly understood by "clock tracers". Clock tracers are (artificial) tracers which increase not only monotonically, but also linearly. Neither SF6 nor CO2 fulfill this criterion. SF6 increases monotonically, but not linearly, CO2 has in addition a seasonal cycle, thus does not even increase monotonically (except if annually averaged). Therefore, the shape of the assumed age spectrum plays a significant role in deriving AoA from such tracers. This is not sufficiently discussed in the manuscript. The authors may want to consult e.g. a recent paper by Fritsch et al. (2020) in ACP on the issue of how AoA can be derived from such tracers and how that agrees with ideal clock tracers. Using such a clock tracer, AoA can be derived as a lag time without needing any knowledge on the age spectrum. While I have not checked all the papers reference on p.2., l. 22, at least Haenel et al. did not use lag time but did take into account the age spectrum. As in the end the main focus is on the comparison of AoA derived in different ways, the use of clear language and correct referencing is necessary. More details are needed on the calculation of AoA, including which tropospheric reference time series have been used, have these been fitted and AoA derived as in Volk et al., (1997)? Or has AoA been derived by convolution of age spectrum and time series? How many years where taken into account in fitting or in convolution etc. Has CO2-production by oxidation of CH4 been taken into account? These are extremely important details which are needed to understand possible discrepancies. Also, I would strongly suggest to include a real clock tracer in the model, from which AoA can then be derived without any assumptions and which can serve as a reference.

[Figure]

*structure of the paper*

The paper presents many aspects, many of which are a repetition of previous work, before finally coming to what is really new, the comparison of CO2 and SF6 reconstructed with the two different methods (BIR and Lagrangian). In my view, many parts of section 2 are not necessary, while other parts are missing. Note that none of the aspects discussed with respect to Figure 1 and 2 are in any way mentioned in the discussion, conclusion or summary. Missing parts are details about the observations and how AoA has been derived from them, but also explanation of methods, e.g. the BIR methods should be explained in brief. Section 2 also is called "Model and Experiment", so I was expecting the usual explanations of which model has been used in which set-up and details about the observations. As it stands now, it is a mixture of model description and interpretation, but does not have any experimental part at all.

**Specific comments**

p.2.l.6: I think that this is a very unlucky formulation and explanation of AoA, as it suggests that an air parcel keeps its integrity during transport.

p.2.l. 9: see discussion above: SF6 and CO2 are not clock-tracers.

p.2.l 13: Note that the results of Engel et al (2009) have been updated in Engel et al., 2017 and Fritsch et al. 2020. (both in ACP)

p.2.l.22. see discussion above: AoA cannot be derived from SF6 or CO2 using the lag-time approach. While this may have been done in the early years of AoA it is certainly not applied in more recent studies.

p.2.l.25: a clock tracer must increase linearly, not only monotonically.

p.3.l.15.: this is only about models, not about experiments. It should include some basic information about the measurements.

p.3.l.26: can this evaluation be summarized?

p.4.l.13: I do not understand this sentence.

p.4.l30-p5.l.13: Is this necessary to understand the rest of the paper?

p.5.l.14: this section should have an introduction to what BIR is.

p.6.l.7: I suggest to use larger AoA, not longer.

p.6.l10: I find it hard to understand this conclusion from the statements above.

p.7.l.11: I find this contradictory: doing a reasonable job enables to do a quantitative assessment?

p.7.l.21: please explain the choice of $\text{Tr}_{trop}$ of 355 K. This is quite a bit below the tropical tropopause and transport from 355 K to the tropical tropopause should still take at least several weeks to months.

p.8.l.1: this is a very large uncertainty range, which is even larger than the central value. Can you explain this large variability and the shape of the distribution (which must be quite unsymmetrical).

p.8.l.6: Delta may not in any way be mistaken as an uncertainty in AoA. It is the width of the spectrum. If you have a perfect tracer, this is completely unrelated to any uncertainty in AoA.

p.8.l.11: I strongly suggest not to call these experimental conditions: these are model parameters used in the investigation.

p.9.l.12: I find it hard to derive this conclusion from the results shown.

p.10.l.20. This means that only about 3

p.10.l.22: This statement can only be made if the cut-off time is included (I suppose 5 years) and is highly dependent on the region in the stratosphere.

p.11.l.5: It has recently been shown that other models have a larger ratio of delta$^2$ to AoA (Hauck et al., 2019, ACP)

p.11.l.14: This statement is not true for clock-tracers, but then as stated above, $CO_2$ and $SF_6$ are no clock tracers.

p.11.l. 23: there are more up to date references for mesospheric loss of $SF_6$, especially Ray et al. (2017, JGR) and Reddmann et al. (2001, JGR).

p.12.l.28: I believe that the authors are wrong: the tail correction is extremely important here, as it has a strong influence on the width of the age spectrum.

p.13.l.6: I am confused: according to Figure A1, AoA is 47 days for sample 03 and delta is 17 days. From this I would derive a ratio of delta$^2$ to AoA of about 6 (and not 0.08).

Fig 9: the x-axis of panel a should not be age. This is transit time.

Fig 10: as before: delta is not in any way a measure of the uncertainty of AoA.

Fig A1: the x-axis of the left hand panels should not be age. This is transit time.

---

## Author Comment (AC1) · 29 Sep 2020

[12pt]article color

**Reply to Referee 1**

We sincerely appreciate Referees 1 and 2 for their review of the manuscript and valuable comments and criticism on it. We understand the problem and have made substantial changes to the manuscript in response to the comments from both Referees. These revisions have significantly improved the manuscript, and we hope we have answered all of the concerns. Our reply to Referee 1 is shown below in blue

[Figure]

following the comments cited in italics.

*In their paper, Nguyen et al. apply two methods (BIR and backward trajectory calculation) to model age of air (spectra) as well as mole fractions of $CO_2$, $SF_6$ and water vapour. In the paper, first the model results are evaluated and subsequently some results are compared with data from a measurement campaign. The comparison works reasonably well, with some discrepancies that mostly can be explained. The overall idea of the paper is good and the method is elaborate. Some of the results are also interesting and can contribute to foster science in this field, although it is for example expectable that chemical $SF_6$ depletion will not allow direct comparison in the upper stratosphere if it is not included in the model. It is good to carve out which of the modelling methods is suitable to tackle which science question.*

*However, the paper is chaotic and does not provide the necessary information to follow. Almost nothing is reported about the measurements of that campaign and what are the points that were supposed to be investigated with them. The model description is unclear, I do not understand why sometimes nudging is described, while the authors apply a CTM, which usually is driven (not nudged) by reanalysis data. That is very confusing. The evaluation of the results is pretty lengthy and should be reduced to about two figures. If need be, the rest can be banished to a supplement (maybe together with the appendix). I like the idea of explaining measurements with modelled AoA spectra, but at the end, that is only a minor part in the paper and is only partly successful (partly due to the sinks). However, my main point really is that the study does not follow a clear research question. The reader can be lost due to that. What is it exactly that is puzzling you about the measurements? Why do you think the applied method can help to answer that question and how do you plan on pursuing that? How can additional information about transport processes be gained through that? What is your contribution to improve the understanding of the underlying processes at the end and how does that fit into existing literature? Some of this information information is*
*lacking, some is spread somewhere across the manuscript, and the reader has to put the pieces together. Further, so many different points are applied, AoA, its spectra, the $CO_2$ and $SF_6$ mole fractions as well as water vapour and the cold point are thrown together, but it does not clearly shine through that all these measures are needed to make the conclusions that are drawn at the end. Also, different models, free running, nudged simulations and/or the CTM and the two diagnostic methods, all that leads to confusion and does not help to get to the point. Additionally, intense use of non-intuitive abbreviations complicate reading and also in the results section, many points are thrown together and a clear focus is missing.*

*Hence, I would suggest the authors to completely revise the manuscript and then submit it again. I think the study can help to advance our understanding of stratospheric transport and the methods that can be used to investigate it if it is presented and structured properly. Please start with one or more clear research questions that can be answered with this method and build everything around that. Use only the methods needed, describe them clearly, and then take the reader point by point towards the conclusion. Please also consider my additional comments that I am making below.*

In response to the above comments, the manuscript has been completely revised by setting clear research questions. Before explaining them briefly, let us resolve the confusion on the use of "ACTM" and the application of nudging in the present study. ACTM is an abbreviation of an Atmospheric General Circulation Model (AGCM)-based Chemistry Transport Model (CTM), which was used in the past literature such as Ishijima et al. (2010, JGR, 115, D20308, doi:10.1029/2009JD013322). We would like to maintain the use as the continuation of previous studies. The application of ACTM with data assimilation is motivated by our hope that realistic transport field is better represented than the direct use of (re)analysis field, such as ERA-Interim, in higher temporal resolution. We choose nudging as the simplest way for data assimilation. Nudging is frequently applied for the diagnosis of model performance in AoA studies as can be

[Figure]

seen from Krol et al. (2018, Geosci. Model Dev., 11, 3109–3130). The improvement attained by the use of ACTM with nudging is discussed in Section 4.

We agree to the comment that the study must follow a clear research question. We have revised the manuscript to state it clearly in Introduction. That is, how can we interpret the vertical profiles of $CO_2$ and $SF_6$ ages obtained by cryogenic air sampling in CUBE/Biak campaign, especially from the aspect the shape of age spectra which is often parameterized by $\Delta^2/\Gamma$ ratio since Hall and Plumb (1994). We employ two methods, boundary impulse response (BIR) method and Lagrangian backward trajectories, both replying on ACTM wind field. The whole manuscript has been revised along the line to answer the question under a unified story. The description on the evaluation of the results and the number of figures are reduced. Due to the additional description such as the measurements of that campaign, however, the total length of the manuscript remains almost the same. Some of the contents are moved to appendices and supplementary material following the suggestion. The use of abbreviation has also been revised. The details are given below.

The manuscript has been reorganized as follows:

1. Introduction

   Our research questions are stated clearly to meet the comments from Referee 1. Recent publications in related topics are also added. Some more descriptions on our campaign CUBE/Biak have been given as well. An introduction of the atmospheric general circulation model-based chemistry transport model is made with its abbreviation ACTM. The use of "clock tracers" is eliminated in response to the comments by Referee 2.

2. Model experiments

   2.1 Description of the model and simulation design

We try to interpret the vertical profiles of observationally estimated $CO_2$- and $SF_6$-ages referring to transport model calculations. The use of ACTM is thus a key to our analysis. Explanation on the use of ACTM is given here.

**2.2 Evaluation of the model performance**

Our results deeply rely on the performance of the transport model. The model performance is briefly investigated by looking at the distribution of tracers that are released as a "pulse" at the tropical surface. This method of tracer release constitutes the basis of the BIR method.

**2.3 Estimation of age spectra and mean age of air**

We employ BIR method and back trajectories to estimate age spectra and mean age of air in the stratosphere. A brief review of the theoretical foundation of both methods are given here before their application to the tropical stratosphere.

**3. Application to CUBE/Biak observations**

**3.1 BIR method**

The mean age estimation relies on unobservable age spectrum. The age spectra estimated from BIR method are described.

**3.2 Lagrangian method**

Back trajectory calculations are often conducted to describe the tracer transport from a Lagrangian point of view. The method is one of the important tools to study stratospheric tracers including water vapor. The use of one-hour averaged one-hour interval wind field, together with additional pressure levels assigned near the tropical tropopause, proved useful to better reproduce the observed profiles of $CO_2$, $SF_6$, and water vapor "tape recorder."

**3.3 Assessment of the mean age profiles**

The mean age profiles derived by applying above two methods are compared against those estimated by using observed $CO_2$ and $SF_6$ mole fractions.

**4. Discussion**

The results obtained above are discussed focusing on the interpretation of the differences between the ACTM-derived and observationally estimated mean ages, $\Delta^2/\Gamma$-ratio and the shape of age spectra, and the advantage of using one-hour averaged one-hour interval data available from ACTM in trajectory calculation.

**5. Summary**

The overall results are summarized.

**Appendix A: Supplementary notes on the age spectra**

The effect of tail correction and fine structure reflecting the pathway difference are discussed emphasizing the importance of using accurate age spectrum for mean age estimation.

**Appendix B: The effect of quasi-biennial oscillation (QBO)**

The modulation of BIR map over the equator due to QBO is briefly described.

Figures are rearranged and reorganized as follows:

**Section 2**

Fig. 1: Latitude-height section of the mixing ratio of January-released pulse tracers in (a) February of the first year, (b) February of the second year, and evolution of pulse tracer concentrations (c) over the equator and (d) at some representative latitudes on 50 hPa pressure surface. Panels (a) and (b) come from original Fig. 1, and panel (c) comes from the upper panels of original Fig. 3. Panel (d) consists of lower panels of original Fig. 3. The original Fig. 2 is deleted.

Fig. 2: Zonal mean distribution of three-year averaged mean age in NH winter (DJF) and summer (JJA). This comes from original Fig. 5. Original Fig. 4 goes to Fig. A1 in Appendix A.

Section 3

Fig. 3: (a) BIR map at 50 hPa over the equator, and (b) latitude-height section of the mean age in March 2015. Panel (a) comes from original Fig. 6 (a), while panel (b) is original Fig. 7. Original Fig. 6 (b) goes to Fig. B1 in Appendix B.

Fig. 4: Age spectra derived from BIR method corresponding to the altitudes of eight cryogenic air samples acquired during CUBE/Biak 2015. This is the same as original Fig. 8.

Fig. 5: Examples of (a) age spectrum and (b) water mixing ratio spectrum estimated from back trajectory method. These panels come from original Fig. 9 (a), (b). Those of original Fig. 9 (c), (d) are deleted.

Fig. 6: Vertical profiles of mole fractions of (a) $CO_2$ (ppm), (b) $SF_6$ (ppt), and (c) water vapor mixing ratio (ppmv) estimated by back trajectories. This figure comes from original Fig. 10. Original Fig. 11 appears in snapshots in a movie provided by Supplementary Material.

Section 4

Fig. 7: Comparison of the vertical profiles of (a) mean age and (b) ratio of moments ($\Delta^2/\Gamma$) estimated by the BIR method, back trajectories, and cryogenic samples. Panel (a) comes from original Fig. 13 after removing horizontal bars for $\Gamma_{\mathrm{bir}}$ and $\Gamma_{\mathrm{trj}}$. Panel (b) is newly plotted from Table 2.

Fig. 8: Time series of the zonal ($u$), meridional ($v$), and vertical ($\omega$) wind components at grid points $0°$ longitude near the equator. This figure comes from original Fig. 12.

Appendix A

Fig. A1: Multi-year averaged age spectra with tail correction estimated by BIR method. This comes from the original Fig. 4.

Fig. A2: (Left) age spectra and (right) meridional projection of back trajectories. This comes from the original Fig. A1.

Appendix B

Fig. B1: A time-height section of mean zonal wind over the equator. This comes from the original Fig. 6 (b).

A supplementary material has been attached with the revised manuscript. It contains an animated GIF showing a meridional projection of air parcels associated with the backward trajectory calculations for one year since the initialization on 27 February 2015.

We believe that the application of two independent methods, BIR and back trajectories, to the ACTM wind field successfully achieved our research goal of interpreting the vertical profiles of $CO_2$ and $SF_6$ ages obtained by cryogenic air sampling in CUBE/Biak campaign. We hope we have made necessary revisions so that the manuscript has reached the required quality for publication in ACP. Detailed revisions associated with Additional comments follow.

**Additional comments:**

P1, L2: *A CTM is not nudged! A CTM uses some meteorological fields for describing transport. This is totally confusing and it is not clear to me what is actually done in this study, because later also you talk about GCM and CTM. Please clarify what that is and what you do throughout the paper.*

As was mentioned at the beginning of our reply, we use an Atmospheric general circulation model (AGCM)-based Chemistry Transport Model (CTM), which is named "ACTM" in previous publications. ACTM employs nudging to reproduce realistic transport for atmospheric chemical/non-chemical components in model.

P1, L4: *Change "a single" to "the chemistry transport model"*

We have changed "a single model" to "the ACTM."

P1, L3–5: *The sentence is unclear. Are there discrepancies between the two models or between models and observations? And the following sentence starts with a "This", but it is unclear what the "this" refers to, to the usefulness, or to the discrepancies.*

This sentence was rewritten together with the following sentence without using the word "discrepancies" and "This.": "Since the BIR method is capable of taking unresolved diffusive processes into account, while the Lagrangian method can distinguish the pathways the air parcels took before reaching the sample site, the application of the two methods to the common transport field simulated by the ACTM is useful in assessing the $CO_2$ and $SF_6$ derived mean ages."

P1, L7: *But where is the connection between the water vapour tape recorder and the mean age here?*

This sentence is revised saying that the advantage is "The capability to examine the reproducibility of the observed values of $CO_2$, $SF_6$, and water vapour".

P1, L8: *Change "the reality" to "good quality" or something alike.*

The phrase "confirming the reality of the trajectory calculations" is deleted.

P2, L12: *Please consider also the newer publication by Engel et al. from 2017 (10.5194/acp–17–6825–2017)*

Thank you for the suggestion. We have updated and cited the suggested paper in the revised manuscript.

P2, L18: *observations and models*

The sentence is revised as "to resolve this discrepancy by reducing uncertainties in both observational and model estimates".

P2, L18: *With "sampling of clock tracers" do you mean $SF_6$ and $CO_2$? Or more tracers? Can you elaborate a little more on the campaign, please, like what, how, how long...?*

Yes, they are $CO_2$ and $SF_6$. However, we revised the manuscript distinguishing $CO_2$ and $SF_6$ from ideal "clock tracers" following the comments from Referee 2. The sentence is deleted and a brief description about CUBE/Biak campaign is added.

P2, L21: *What exactly is it that is puzzling you about these measurements? This should be central in all parts of the paper.*

We are sorry that we did not explain well about our research question. As was mentioned in our reply to your major comments above, our research question is clearly written in Introduction.

P2, L25: *But that is why the Green's function is used to flatten out these non-linearities. Please see and possibly mention Fristch et al. 2020 (10.5194/acp–2019–974) and citations therein.*

Thank you for the suggestion. We have added the suggested paper for a discussion in the revised manuscript.

P3, L17–18: *So is it a CTM or a GCM now?*

We are sorry for the confusion, but we do hope this question has been already resolved from our reply at the beginning.

P3, L23: *It would be easier to use the name of the model from here on, instead of "ACTM", if the model has a name.*

Again we hope this question has been resolved already; "ACTM" is the name of the model used in this study.

P3, L27: *How did the model perform in that inter-comparison? Was it somewhere around the multi-model mean or was it an outlier?*

As the ACTM was nudged to JRA-25 (not to ERA-Interim) in the inter-comparison by Krol et al. (2018), the results need to be interpreted carefully. They found that ACTM showed the strongest convective mixing in the tropics and the youngest air at the high-altitude poles among the models participated in the comparison. This is stated in Sect. 2.3.

P4, L5: *"several years". Please be precise, for the sake of reproducibility. Did you use ERA-I data of year 2004 for that and repeat that year for ? years?*

Our simulation has been conducted for the period from 1 January 2000 to 31 March 2015 by nudging horizontal winds and temperature to ERA-Interim data. The first five years (January 2000 to December 2004) are regarded as the spin-up period. This information is given in Sect. 2.1.

*Sects. 2.3 and 2.4 are just the evaluation of the model. Firstly, this should be reflected in the section headers, and secondly, these sections should be considerably shortened and/or moved partly to the supplement. I suggest to reduce the number of figures from 5 to 2.*

Thank you for your suggestions. The whole Sect. 2 has been rewritten by reducing the number of figures to 2. Fig. 4 is moved to Appendix for the sake of readability. Sect. 2.2 is entitled "Evaluation of the model performance" following the suggestion.

*These abbreviations, particularly "AF" and "AN" do not make much sense to me.*

We are sorry for the confusion but these follow our precedent use in BAMS paper. For clarity and readability, those abbreviations have been changed to ACTM-FREE (for AF) and ACTM-NUDG (for AN). Additionally, "EI" is also changed to "ERA-Interim".

*Fig. 2: I do not fancy that the streamfunction is shown again here, it was shown in Fig. 1 already and does not help much, instead, it disturbs the view on the tendencies. I suggest to remove it.*

Thank you for the comment. As suggested by the reviewer, this figure has been removed.

P5, L15: *What do you mean by "selected tracers"?*

The term "selected tracers" was intended to identify pulse tracers released at a specific month such as January. The term is no longer used in revised manuscript: "The transport features described above are limited to those at a specific time in the Northern winter." (Sect. 2.2)

P7, L1: *...temperature move upward over time.*

The sentence has been removed associated with the revision of the paragraph.

[Figure]

P7, L6: *backward*

Done.

P7, L15–17: *Unclear and awkward phrasing, please rephrase. Plus, what is meant by "not simple"?*

The sentence has been rephrased to the following without using "not simple": "Therefore, it is necessary to address the difference in the definition of the reference time from which age counting is started before making direct comparison between the two. As is evident from Fig. 1, the excursion of the tropospheric air to the stratosphere depends on tropospheric transport features, including isentropic mixing with the air in the extratropical LS, and thus the mean age counted from the tropical surface is not always a sum of the tropospheric residence time and the mean age counted from the TTL."

P7, L25–26: *Remove parentheses around dates*

Thank you for the comment. The parentheses are used to identify two-dimensional coordinates on the BIR map in the form (source time, field time). Thus, we would like to retain the parentheses around dates with the following modification: "at $(t', t)$ = (March 2007, October 2007) and (November 2009, July 2010)."

P7, L28: *change "drives the tracers upward" to "intensifies the upward tracer transport"*

This sentence is moved to Appendix B and rephrased to: "the upward tracer transport driven by extratropical pumping is intensified by the secondary circulation"

P7, L31: *"The vertical axis is ...." What is that sentence supposed to mean? It makes no sense to me.*

This sentence has been removed.

P7, L32: *What is a latitudinal split? Please explain clearly what you talk about. Moreover, why is that important now, you explained the QBO topic already before.*

Again, we are sorry for the confusion. The sentence has been rephrased to: "The deformation of the contours at 3.0, 3.5, and 4.0 years showing wavy structures in the tropics are due to the downward motion associated with the westerly shear of the QBO (Appendix B)." (Sect. 3.1)

P8, L5: *...the spread of the transit times ($\Delta$)...*

"($\Delta$)" has been inserted as suggested.

P8, L9: *A bundle! Please be more quantitative.*

The sentence has been revised referring to Table 1 for the details of the model parameters of trajectory calculations.

P8, L9: *the spectra of AoA, $CO_2$ and $SF_6$ mole fractions? You are mixing up something here, please be specific.*

In this work, the trajectories are used to estimate not only the spectra of stratospheric AoA, but also the $CO_2$ and $SF_6$ mole fractions as well as the water vapor mixing ratio by tracking the position of air parcels advected by the 3D wind. The sentence and panels (c) and (d) of Fig. 9 are deleted following the revisions of the manuscript.

P8, L11: *Can you still give a very brief description of the method of analysis please.*

Some descriptions are made: "In the Lagrangian method, the age spectra are estimated by counting the transit time $\tau$ during the advection along each kinematic trajectory since the last passage through the top of the troposphere ($Tr_{top}$)." (Sect. 2.3) and "The present study tries to resolve disagreements between the

estimates from trajectory calculations and CUBE/Biak observations by increasing the number of trajectories and extending the integration period (Table 1)." (Sect. 3.2).

P8, L12–14: *I do not understand what this sentence is supposed to mean. Please rephrase it and sharpen the message.*

It is an important advantage of using the ACTM and is rephrased to: "In addition to using the ERA-Interim analysis directly as 6-hour interval snapshots, the assimilated meteorological field created by nudging its horizontal winds and temperature to the ACTM are also used for trajectory calculations. In this case, one-hour averaged values are used at one-hour intervals (ACTM-NUDG)."

P8, L14: *Where do these additional levels come from?*

Additional pressure levels are set to better represent the Lagrangian cold-point temperature that controls the water transport to the stratosphere. All pressure-level data, not restricted to the additional levels, are interpolated from model level data.

P8, L21: *What is CONTRAIL data? Please describe!*

The citation of CONTRAIL data is deleted as the name is not absolutely important. The description is revised to: "The tropospheric reference was derived from direct measurements of air samples collected onboard commercial airliners during the cruise within the area $5°$ S–$5°$ N and $142°$ E–$150°$ E at an altitude of 10-13 km".

*Fig. 9 Water "vapour"! Or is ice included too? (Throughout the paper!)*

Water vapour is an important constituent to describe the ascending motion in the equatorial stratosphere, although total hydrogen (=$H_2O$ + 2$CH_4$) is a better quantity (Waugh and Hall, 2002, Rev. Geophys.). Trajectory calculations are

frequently used to estimate stratospheric water variations (e.g., Fueglistaler and Haynes, 2005, JGR). The top right panel for water vapour is retained as the right panel of Fig. 5 with description in Sect. 3.2. We have never mentioned ice.

P9, L2 and L5: *You already defined these abbreviations above.*

Thank you for the comment. We have deleted the redundant information.

P9, L3: *But what were the problems in H18? Can you provide a quick introduction? Without that, it is almost impossible to follow.*

The following descriptions are given in Sect. 3,2: "The present study tries to resolve disagreements between the estimates from trajectory calculations and CUBE/Biak observations by increasing the number of trajectories and extending the integration period (Table 1). Improvements are not limited to these simulation settings. In addition to using the ERA-Interim analysis directly as 6-hour interval snapshots, the assimilated meteorological field created by nudging its horizontal winds and temperature to the ACTM are also used for trajectory calculations. In this case, one-hour averaged values are used at one-hour intervals (ACTM-NUDG)."

P9, L18: *"It is interesting". Does that mean the other results are not interesting?*

Thank you for the comment. Figure 11 is replaced by a movie provided as a Supplementary Material. Related descriptions are made in Sect. 4: "The time evolution of the meridional location of air parcels corresponding to Sample 8 is visualized as a movie in Supplementary Material. We can see that air parcels gradually descend in reverse time sequence, and stay mostly inside the "tropical pipe" without appreciable latitudinal dispersion during the Northern winter (January and February 2015). The vertical advection is fastest in ACTM-FREE and slowest in ACTM-NUDG. By June 2014, an appreciable number of ACTM-FREE air parcels has descended back into the troposphere. In contrast, scarcely any

air parcels are found in the troposphere in ACTM-NUDG. ERA-Interim shows features intermediate between the two."

P9, L18–32: *Using these abbreviations this way make it almost impossible to follow.*

The abbreviations "EI", "AN", and "AF" follow the use in our project paper (H18). Now we have changed AN to ACTM-NUDG, AF to ACTM-FREE and EI to ERA-Interim.

P9, L25: *How is that question linked with the general idea of the paper?*

Our purpose is to attain a better understanding on the vertical profiles of $CO_2$ and $SF_6$ ages obtained by cryogenic air sampling in CUBE/Biak campaign. We tried to reproduce the observed age profiles by applying the BIR and Lagrangian methods to the meteorological fields simulated by ACTM. The question, "Why is the advection velocity in EI and AN different in spite of nudging?" is linked to the reproducibility of the observations. It is natural for us to expect that AN (ACTM-NUDG) gives similar results to EI (ERA-Interim), as AN is nudged to EI. The related descriptions are made in Sect. 4 as described in our reply to your comment on P9, L18.

P10, L5–9: *Even more abbreviations that totally disturb readability.*

Following this comment, we stoped using $\Gamma^*$ and $\Gamma_{adj}$ in the revised manuscript. However, we want to retain the use of $\Gamma_{corr}$, $\Gamma_{bir}$, $\Gamma_{trj}$, $\Gamma_{Cobs}$, and $\Gamma_{Sobs}$ for better readability of the manuscript. We don't think it wise to write, for example, "mean age estimated by BIR method having been applied the tail correction and source-region adjustment" every time for $\Gamma_{bir}$.

---

## Author Comment (AC2) · 29 Sep 2020

[12pt]article color

**Reply to Referee 2**

We sincerely appreciate Referees 1 and 2 for their review of the manuscript and valuable comments and criticism on it. We understand the problem and have made substantial changes to the manuscript in response to the comments from both Referees. These revisions have significantly improved the manuscript, and we hope we have answered all of the concerns. Our reply to Referee 2 is shown below in blue

[Figure]

following the comments cited in italics.

*The paper by Nguyen et al. investigates different methods with which observations of mean age tracers and mean age of air (AoA) can be reconstructed. While the study is certainly in the scope of ACP and the subject is of significant scientific interest, there are a number of problems I see in the investigations presented here. In my view, the authors misinterpret some aspects of AoA, as detailed below. Further, in my view some aspects are presented (e.g. related to Figures 1 and 2) which are not taken up in the discussion or conclusions of the manuscript and are not necessary for the understanding. Other aspects like experimental details (e.g. where, when and how were the samples taken) and how was AoA calculated from the observations is omitted. There are also a number of important recent papers, which are not included in the discussion (see details given below). Due to these issues, I believe that the paper is not ready for publication, but needs major revisions before it can be considered for publication.*

We really appreciate detailed review and valuable comments to the manuscript. We believe that the application of two independent methods, the boundary impulse response (BIR) method and back trajectories, to the ACTM wind field successfully achieved our research goal of interpreting the vertical profiles of $CO_2$ and $SF_6$ ages obtained by cryogenic air sampling in CUBE/Biak campaign. We think our revisions detailed below are enough to satisfy the reviewer and hopefully make the manuscript suitable for publication in ACP.

**Major comments**

Clock tracers and derivation of AoA

*I believe that there is a misinterpretation on what is commonly understood by "clock tracers". Clock tracers are (artificial) tracers which increase not only monotonically, but*

*also linearly. Neither SF$_6$ nor CO$_2$ fulfill this criterion. SF$_6$ increases monotonically, but not linearly, CO$_2$ has in addition a seasonal cycle, thus does not even increase monotonically (except if annually averaged). Therefore, the shape of the assumed age spectrum plays a significant role in deriving AoA from such tracers. This is not sufficiently discussed in the manuscript. The authors may want to consult e.g. a recent paper by Fritsch et al. (2020) in ACP on the issue of how AoA can be derived from such tracers and how that agrees with ideal clock tracers. Using such a clock tracer, AoA can be derived as a lag time without needing any knowledge on the age spectrum. While I have not checked all the papers reference on p.2., l. 22, at least Haenel et al. did not use lag time but did take into account the age spectrum. As in the end the main focus is on the comparison of AoA derived in different ways, the use of clear language and correct referencing is necessary. More details are needed on the calculation of AoA, including which tropospheric reference time series have been used, have these been fitted and AoA derived as in Volk et al., (1997)? Or has AoA been derived by convolution of age spectrum and time series? How many years where taken into account in fitting or in convolution etc. Has CO$_2$-production by oxidation of CH$_4$ been taken into account? These are extremely important details which are needed to understand possible discrepancies. Also, I would strongly suggest to include a real clock tracer in the model, from which AoA can then be derived without any assumptions and which can serve as a reference.*

Appreciating that CO$_2$ and SF$_6$ are conveniently used to visualize stratospheric general circulation under the scope of pseudo-"clock tracers," we understand it is necessary to distinguish them from the idealized concept of "clock tracers." The phrases such as "observations of clock tracer" are replaced by "observations of tracer", for example, wherever necessary. We also understand the importance of age spectra to estimate mean AoA from tracer observations. In the present study, the BIR method and the back trajectory calculations are applied to estimate age spectra and they are used to calculate the mean age by convolution. We understand the descriptions were

not sufficient, and revised the manuscript by adding more explanation. As for additional model experiments that use a real clock tracer, we should say it is not possible in the foreseeable future because of the lack of computing resources; the model experiments presented in this study were conducted when one of the coauthors (KI) was affiliated in JAMSTEC before his move to MRI. Following the above comments, description on our field experiments and the method of mean age estimation from the observed $CO_2$ and $SF_6$ mole fractions intensionally omitted referring to our publications (Hasebe et al. 2018, Sugawara et al. 2018), are made in some details. All cited references are rechecked and revised based on the reviewer's comments.

Structure of the paper

*The paper presents many aspects, many of which are a repetition of previous work, before finally coming to what is really new, the comparison of $CO_2$ and $SF_6$ reconstructed with the two different methods (BIR and Lagrangian). In my view, many parts of section 2 are not necessary, while other parts are missing. Note that none of the aspects discussed with respect to Figure 1 and 2 are in any way mentioned in the discussion, conclusion or summary. Missing parts are details about the observations and how AoA has been derived from them, but also explanation of methods, e.g. the BIR methods should be explained in brief. Section 2 also is called "Model and Experiment", so I was expecting the usual explanations of which model has been used in which set-up and details about the observations. As it stands now, it is a mixture of model description and interpretation, but does not have any experimental part at all.*

The manuscript has been reorganized as follows:

1. Introduction
    Our research questions are stated clearly to meet the comments from Referee 1. Recent publications in related topics are also added. Some more descriptions

on our campaign CUBE/Biak have been given as well. An introduction of the atmospheric general circulation model-based chemistry transport model is made with its abbreviation ACTM. The use of "clock tracers" is eliminated in response to the comments by Referee 2.

2. Model experiments

2.1 Description of the model and simulation design
We try to interpret the vertical profiles of observationally estimated $CO_2$- and $SF_6$-ages referring to transport model calculations. The use of ACTM is thus a key to our analysis. Explanation on the use of ACTM is given here.

2.2 Evaluation of the model performance
Our results deeply rely on the performance of the transport model. The model performance is briefly investigated by looking at the distribution of tracers that are released as a "pulse" at the tropical surface. This method of tracer release constitutes the basis of the BIR method.

2.3 Estimation of age spectra and mean age of air
We employ BIR method and back trajectories to estimate age spectra and mean age of air in the stratosphere. A brief review of the theoretical foundation of both methods are given here before their application to the tropical stratosphere.

3. Application to CUBE/Biak observations

3.1 BIR method
The mean age estimation relies on unobservable age spectrum. The age spectra estimated from BIR method are described.

3.2 Lagrangian method
Back trajectory calculations are often conducted to describe the tracer transport from a Lagrangian point of view. The method is one of the important

tools to study stratospheric tracers including water vapor. The use of one-hour averaged one-hour interval wind field, together with additional pressure levels assigned near the tropical tropopause, proved useful to better reproduce the observed profiles of $CO_2$, $SF_6$, and water vapor "tape recorder."

**3.3 Assessment of the mean age profiles**

The mean age profiles derived by applying above two methods are compared against those estimated by using observed $CO_2$ and $SF_6$ mole fractions.

**4. Discussion**

The results obtained above are discussed focusing on the interpretation of the differences between the ACTM-derived and observationally estimated mean ages, $\Delta^2/\Gamma$-ratio and the shape of age spectra, and the advantage of using one-hour averaged one-hour interval data available from ACTM in trajectory calculation.

**5. Summary**

The overall results are summarized.

**Appendix A: Supplementary notes on the age spectra**

The effect of tail correction and fine structure reflecting the pathway difference are discussed emphasizing the importance of using accurate age spectrum for mean age estimation.

**Appendix B: The effect of quasi-biennial oscillation (QBO)**

The modulation of BIR map over the equator due to QBO is briefly described.

Figures are rearranged and reorganized as follows:

Section 2

Fig. 1: Latitude-height section of the mixing ratio of January-released pulse tracers in (a) February of the first year, (b) February of the second year, and evolution of pulse tracer concentrations (c) over the equator and (d) at some representative latitudes on 50 hPa pressure surface. Panels (a) and (b) come from original Fig. 1, and panel (c) comes from the upper panels of original Fig. 3. Panel (d) consists of lower panels of original Fig. 3. The original Fig. 2 is deleted.

Fig. 2: Zonal mean distribution of three-year averaged mean age in NH winter (DJF) and summer (JJA). This comes from original Fig. 5. Original Fig. 4 goes to Fig. A1 in Appendix A.

Section 3

Fig. 3: (a) BIR map at 50 hPa over the equator, and (b) latitude-height section of the mean age in March 2015. Panel (a) comes from original Fig. 6 (a), while panel (b) is original Fig. 7. Original Fig. 6 (b) goes to Fig. B1 in Appendix B.

Fig. 4: Age spectra derived from BIR method corresponding to the altitudes of eight cryogenic air samples acquired during CUBE/Biak 2015. This is the same as original Fig. 8.

Fig. 5: Examples of (a) age spectrum and (b) water mixing ratio spectrum estimated from back trajectory method. These panels come from original Fig. 9 (a), (b). Those of original Fig. 9 (c), (d) are deleted.

Fig. 6: Vertical profiles of mole fractions of (a) $CO_2$ (ppm), (b) $SF_6$ (ppt), and (c) water vapor mixing ratio (ppmv) estimated by back trajectories. This figure comes from original Fig. 10. Original Fig. 11 appears in snapshots in a movie provided by Supplementary Material.

Section 4

Fig. 7: Comparison of the vertical profiles of (a) mean age and (b) ratio of moments ($\Delta^2/\Gamma$) estimated by the BIR method, back trajectories, and cryogenic samples. Panel (a) comes from original Fig. 13 after removing horizontal bars for $\Gamma_{bir}$ and $\Gamma_{trj}$. Panel (b) is newly plotted from Table 2.

Fig. 8: Time series of the zonal ($u$), meridional ($v$), and vertical ($\omega$) wind components at grid points $0°$ longitude near the equator. This figure comes from original Fig. 12.

Appendix A

Fig. A1: Multi-year averaged age spectra with tail correction estimated by BIR method. This comes from the original Fig. 4.

Fig. A2: (Left) age spectra and (right) meridional projection of back trajectories. This comes from the original Fig. A1.

Appendix B

Fig. B1: A time-height section of mean zonal wind over the equator. This comes from the original Fig. 6 (b).

A supplementary material has been attached with the revised manuscript. It contains an animated GIF showing a meridional projection of air parcels associated with the backward trajectory calculations for one year since the initialization on 27 February 2015.

**Specific comments**

P2, L6: *I think that this is a very unlucky formulation and explanation of AoA, as it suggests that an air parcel keeps its integrity during transport.*

We are afraid we may not understand you correctly, but Kida (1983, JMSJ, 61, 517) is referred by many studies such as Hall and Plumb (1994) and Waugh and Hall (2002), and we understand his schematic illustration (Fig. 4) is well-accepted as a basic concept in our community. The sentence is retained as it is.

P2, L9: *see discussion above: SF$_6$ and CO$_2$ are not clock-tracers.*

As is mentioned in the reply on Major commnets, we no longer call SF$_6$ and CO$_2$ as clock tracers in the whole manuscript.

P2, L13: *Note that the results of Engel et al. (2009) have been updated in Engel et al., 2017 and Fritsch et al. 2020. (both in ACP)*

Thank you for the comment. Both papers are cited in Sect. 4.

P2, L22: *see discussion above: AoA cannot be derived from SF$_6$ or CO$_2$ using the lag-time approach. While this may have been done in the early years of AoA it is certainly not applied in more recent studies.*

We agree with the referee and the lag-time method is not used in our study. This sentence has been removed along with the revisions.

P2, L25: *a clock tracer must increase linearly, not only monotonically.*

We agree with the referee. This sentence is deleted along with the revisions.

P3, L15: *this is only about models, not about experiments. It should include some basic information about the measurements.*

Section 2 is now entitled "Model experiments" and revised to include description of the model and simulation design (Sect. 2.1), evaluation of the model performance (Sect. 2.2), and estimation of age spectra and mean age of air (Sect. 2.3). The method of estimating mean age from air samples is discussed in the second paragraph of Sect. 4 (Discussion), although we do not describe anything on the

measurements such as how to collect air samples and how to measure mole fraction of $CO_2$ from them.

P3, L26: *can this evaluation be summarized?*

The evaluation of our model performance is made in Sect. 2.2. The results by Krol et al. (2018) are briefly introduced in Sect. 2.3.

P4, L13: *I do not understand this sentence.*

The sentence is rephrased to: "The Northern winter transport field is investigated as an example in the top panels of Fig. 1 by examining the zonal-mean distribution of January-released tracers in (a) February of the first year (i.e., the next month) and (b) February of the second year (i.e., 13th month since release)." in Sect. 2.2.

P4, L30 – P5, L13: *Is this necessary to understand the rest of the paper?*

Thank you for the comment. This part is totally revised into new Sect. 2.2, along with the elimination of Fig. 2 and associated descriptions on the transport features.

P5, L14: *this section should have an introduction to what BIR is.*

Section 2.4 is totally rewritten following your comment. The revised Sect. 2.3 includes a brief introduction on the theoretical foundation of the BIR method.

P6, L7: *I suggest to use larger AoA, not longer.*

We have changed "longer" to "older" referring to preceding studies such as Li et al. (2012a, b) and Ploeger and Birner (2016).

P6, L10: *I find it hard to understand this conclusion from the statements above.*

We understand your concern. This sentence is deleted, and the whole paragraph including Fig. 4 is moved to Appendix A.

P7, L11: *I find this contradictory: doing a reasonable job enables to do a quantitative assessment?*

We think a quantitative assessment can be made only by a good job. Anyway, the sentence is deleted.

P7, L21: *please explain the choice of $Tr_{top}$ of 355 K. This is quite a bit below the tropical tropopause and transport from 355 K to the tropical tropopause should still take at least several weeks to months.*

The 355 K isentrope is surely much lower than the tropical tropopause. On the other hand, it is close to the altitude of tropospheric reference adopted by Sugawara et al. (2018). The meterological reason of the choice is written in Sect. 2.3 as follows: "$Tr_{top}$ is taken to be the 355 K isentropic surface, reflecting the fact that the influence of tropical convective motion almost ceases at this level and diabatic forcing gradually changes to radiative heating in and above the TTL (Hasebe and Noguchi, 2016)."

P8, L1: *this is a very large uncertainty range, which is even larger than the central value. Can you explain this large variability and the shape of the distribution (which must be quite unsymmetrical).*

We are really grateful to this comment. There was a mistake in our calculation, and the correct number is 0.24 years rather than 0.69 years.

P8, L6: *Delta may not in any way be mistaken as an uncertainty in AoA. It is the width of the spectrum. If you have a perfect tracer, this is completely unrelated to any uncertainty in AoA.*

We understood and totally agree with the referee about this. Figure 8 is now Fig. 4 in the revised manuscript, and the phrase "as the estimates of uncertainties in $\Gamma_{corr}$" is deleted.

P8, L11: *I strongly suggest not to call these experimental conditions: these are model parameters used in the investigation.*

The sentence is changed to: "The model parameters for calculating the kinematic backward trajectories are summarized in Table 1".

P9, L12: *I find it hard to derive this conclusion from the results shown.*

The revised sentence reads, "A remarkable improvement in the use of ACTM-NUDG on H18 and the insufficient performance of ACTM-FREE are further discussed in Sect. 4 investigating the vertical mass transport." (Sect. 3.2). Figure 11 is replaced by a movie to be provided as Supplementary Material.

P10, L20: *This means that only about 3*

We are afraid we do not understand this comment correctly, but we expect 3 years will be an overestimation for the mean age around 25–30 km considering the perceived overestimation of $\Gamma_{\text{Sobs}}$ arising from the mesospheric loss of $SF_6$. In any case, we would like to focus our evaluation of the observed age profile relative to mean ages obtained by BIR and Lagrangian methods. The sentence is revised to: "The omission of this pathway must result in the underestimation of $\Gamma_{\text{trj}}$ relative to $\Gamma_{\text{bir}}$. It is also responsible for making $\Gamma_{\text{trj}}$ younger than $\Gamma_{\text{Cobs}}$ and $\Gamma_{\text{Sobs}}$ above 25 km. That is, the absence of mesospheric air parcels in the Lagrangian calculations leads to the higher mole fractions (Fig. 6) and the younger mean age (Fig. 7) than the observational values."
Please note that Fig. 13 (now the left panel of Fig. 7) is revised by eliminating horizontal bars for $\Gamma_{\text{trj}}$ and $\Gamma_{\text{bir}}$ because $\Delta$ cannot be used as a measure of uncertainties. The bars for $\Gamma_{\text{Cobs}}$ and $\Gamma_{\text{Sobs}}$ reflect the uncertainties associated with the laboratory analysis to derive $CO_2$ and $SF_6$ mole fractions.

P10, L22: *This statement can only be made if the cut-off time is included (I suppose 5 years) and is highly dependent on the region in the stratosphere.*

[Figure]

The sentence is rephrased to: "while more than 50 % of the mean age comes from the tail when the transit time is cut-off at 4 years in the BIR method applied to the extratropical stratosphere (Li et al., 2012b)."

P11, L5: *It has recently been shown that other models have a larger ratio of delta[2] to AoA (Hauck et al., 2019, ACP)*

Thank you for the introduction of recent result. Hauck et al. (2019) is cited together with Fritsch et al. (2020) in Sect. 4 discussing the value of the ratio of moments. Concerning this value, we found an error in the estimation of the width of age spectrum in the BIR method. The corrected results are updated in the revised manuscript (Table 2). The discussion related to this parameter is also revised.

P11, L14: *This statement is not true for clock-tracers, but then as stated above, $CO_2$ and $SF_6$ are no clock tracers.*

As is mentioned on our reply to Major comments, we no longer call $CO_2$ and $SF_6$ as clock tracers.

P11, L23: *there are more up to date references for mesospheric loss of $SF_6$, especially Ray et al. (2017, JGR) and Reddmann et al. (2001, JGR).*

We appreciate this input. The suggested papers are cited in the revised manuscript.

P12, L28: *I believe that the authors are wrong: the tail correction is extremely important here, as it has a strong influence on the width of the age spectrum.*

Generally speaking, the tail correction is extremely important. However, here in the tropical lower stratosphere at around 17 to 18 km, the mean ages estimated by trajectory calculations are 0.12 to 0.13 years and the tail corrections are on the order of days. The sentence, "The tail correction is negligible and is ignored here." is retained as it is.

P13, L6: *I am confused: according to Figure A1, AoA is 47 days for sample 03 and delta is 17 days. From this I would derive a ratio of delta$^2$ to AoA of about 6 (and not 0.08).*

We are sorry to find incorrect citation of the numbers shown in Fig. A1 (now Fig. A2). Correct values are AoA = 49 days and delta = 32 days for Sample 3. From this the ratio of delta$^2$ to AoA is 20.9 days = 0.06 years. Similarly the ratio is 0.32 years for Sample 1. Table 2 is corrected.

*Fig 9: the x-axis of panel a should not be age. This is transit time.*

Corrected.

*Fig 10: as before: delta is not in any way a measure of the uncertainty of AoA.*

The horizontal bars are deleted together with the associated captions.

*Fig A1: the x-axis of the left hand panels should not be age. This is transit time.*

Corrected (now renumbered to Fig. A2).